# ACCELERATED PREFERENCE OPTIMIZATION FOR LARGE LANGUAGE MODEL ALIGNMENT

## ABSTRACT

Reinforcement Learning from Human Feedback (RLHF) has emerged as a pivotal tool for aligning large language models (LLMs) with human preferences. Direct Preference Optimization (DPO), one of the most popular approaches, formulates RLHF as a policy optimization problem without explicitly estimating the reward function. It overcomes the stability and efficiency issues of two-step approaches, which typically involve first estimating the reward function and then optimizing the policy via proximal policy optimization (PPO). Since RLHF is essentially an optimization problem, and it is well-known that momentum techniques can accelerate optimization both theoretically and empirically, a natural question arises: Can RLHF be accelerated by momentum? This paper answers this question in the affirmative. In detail, we first show that the iterative preference optimization method can be viewed as a proximal point method. Based on this observation, we propose a general Accelerated Preference Optimization (APO) framework, which unifies many existing preference optimization algorithms and employs Nesterov's momentum technique to speed up the alignment of LLMs. Theoretically, we demonstrate that APO can achieve a faster convergence rate than the standard iterative preference optimization methods, including DPO and SPPO. Empirically, we show the superiority of APO over DPO, iterative DPO, and other strong baselines for RLHF on the AlpacaEval 2.0 benchmark.

## 1 INTRODUCTION

Large Language Models (LLMs) have emerged as a pivotal technique in the era of artificial general intelligence and recently demonstrated impressive capabilities in tasks such as text generation (Bubeck et al., 2023; Anil et al., 2023; Touvron et al., 2023), coding (Chen et al., 2021; Austin et al., 2021), and problem solving (Cobbe et al., 2021; Wei et al., 2022). A key element contributing to these achievements is the alignment of LLMs with human preference data, utilizing reinforcement learning from human feedback (RLHF) (Ziegler et al., 2019; Christiano et al., 2017; Ouyang et al., 2022; Bai et al., 2022; Munos et al., 2023).

The standard RLHF method (Ouyang et al., 2022) involves three main steps: feedback collection, reward modeling, and policy optimization. Specifically, the LLM receives human-generated prompts and produces several possible responses. Subsequently, human preferences for these responses are collected and used to train a reward model that matches these preferences. Finally, the policy optimization process updates the large language model using optimization algorithms such as Proximal Policy Optimization (PPO) (Schulman et al., 2017) to produce responses with higher preference based on the trained reward model.

Recently, Rafailov et al. (2023) introduced the Direct Preference Optimization (DPO) method based on the Bradley-Terry (BT) model (Bradley & Terry, 1952). This method skips the reward modeling process and replaces it with a reparameterized reward with respect to the policy, allowing for direct optimization of the LLM. It simplifies the implementation, while maintaining a comparable or even better performance than standard RLHF. Later, several works (Munos et al., 2023; Azar et al., 2024; Wu et al., 2024; Rosset et al., 2024) considered the general preference model rather than the BT model, and designed algorithms (Nash-MD, IPO, SPPO, and DNO) that more flexibly represent human preferences while maintaining a simple implementation. Nonetheless, both the standard RLHF process and its variations rely on the policy optimization process. In fact, for general optimization problems, it is well-known that momentum techniques (Polyak, 1964; Nesterov, 2013;

Bubeck et al., 2015b) can accelerate the convergence of the optimization algorithm in both theory and practice. Therefore, a natural question arises:

Can reinforcement learning from human feedback be accelerated?

In this paper, we affirmatively answer this question. In details, following Xu et al. (2023); Yuan et al. (2024b); Chen et al. (2024); Wu et al. (2024), we consider the iterative preference optimization framework. In this framework, a series of models is constructed, with each model being improved based on the previous one using preference optimization algorithm such as DPO and SPPO. We demonstrate that the policy optimization problem under this framework resembles the proximal point method (Bregman, 1967; Censor & Zenios, 1992; Kiwiel, 1997). Based on this observation, we adapt Nesterov's acceleration method (Nesterov, 1983; 2008; 2013; Lin et al., 2018), and introduce a variant of the iterative preference optimization framework, named Accelerated Preference Optimization (APO). At the core of APO is an extrapolation step after each policy update. The contributions of our work are highlighted as follows:

- We propose a general preference optimization framework, APO, based on Nesterov's momentum to accelerate preference optimization. Our theoretical analysis shows that iterative DPO achieves an $\widetilde{O}(1/t)$ sub-optimality gap from the optimal policy under the Bradley-Terry (BT) model. As a comparison, our algorithm APO achieves a smaller $\widetilde{O}((1-\alpha)/t)$ sub-optimality gap, where $\alpha$ is the extrapolation parameter in the momentum. To the best of our knowledge, our work provides the first convergence analysis of vanilla iterative DPO and the first accelerated preference optimization algorithm with provable guarantees.

- With an additional minimal sub-optimality gap assumption in the BT model, we prove that APO will convergence to the optimal policy in total variation (TV) distance at the rate of $\exp\big(-O(t/(1-\alpha))\big)$, improving the $\exp\big(-O(t)\big)$ rate of iterative DPO. In addition, we extend our results to the general preference model and show that APO with the SPPO loss function can also accelerate SPPO under a similar minimal sub-optimality gap assumption.

- Empirically, we verify the performance of APO when applied to DPO method on fine-tuning `Mistral-7B-Instruct-v0.2`. In the AlpacaEval 2.0 evaluation tasks, APO with 3 iterations achieves a length-controlled win rate of 31.73%, demonstrating a 1.78% improvement over iterative DPO and a 5.34% improvement over Snorkel's `Mistral-PairRM-DPO`. Moreover, APO with 2 iterations obtains a win rate of 37.53%, matching iterative DPO's 37.65% with 3 iterations, with noticeably shorter response lengths. This is consistent with our theoretical analysis. In addition, the evaluation on five general instruction-following tasks from MT-Bench shows an average score of 9.57 out of 10, further demonstrating APO's superior performance.

**Notation.** For any positive integer $n$, we employ $[n]$ to denote the set $\{0, \ldots, n\}$. For two sequences $\{a_n\}$ and $\{b_n\}$, we write $a_n = O(b_n)$ if there exists an absolute constant $C$ such that $a_n \leq Cb_n$. We use $\widetilde{O}(\cdot)$ to further hide the logarithmic factors.

## 2 PRELIMINARIES

In the setting of RLHF, we assume a finite context set $\mathcal{X}$, and possible response set $\mathcal{Y}$. For any prompts $x \in \mathcal{X}$, a policy $\pi : \mathcal{X} \to \Delta(\mathcal{Y})$ maps the prompt $x$ to the discrete distributions over the response set $\mathcal{Y}$. For a given context $x \in \mathcal{X}$ collected from distribution $\rho$, we generate two responses $y_1, y_2$ with a reference policy $\mu$ and receive preferences from either humans or more advanced language models between these two responses $(y^w \succ y^l)$, where $y^w$ and $y^l$ represent the preferred and dispreferred generated responses in $\{y_1, y_2\}$. Following Christiano et al. (2017); Ouyang et al. (2022); Rafailov et al. (2023), we assume the existence of a latent reward model $r^*(x, y)$, and the preference distribution satisfies the Bradley-Terry (BT) model (Bradley & Terry, 1952):

$$P(y_1 \succ y_2|x) = \frac{\exp\big(r^*(x, y_1)\big)}{\exp\big(r^*(x, y_1)\big) + \exp\big(r^*(x, y_2)\big)}. \tag{2.1}$$

Under this assumption, the standard RLHF first estimates the reward model by minimizing the following negative log-likelihood of BT model:

$$\mathcal{L}(r) = -\mathbb{E}_{(x, y^w, y^l) \sim \mathcal{D}}\big[\log \sigma\big(r(x, y^w) - r(x, y^l)\big)\big], \tag{2.2}$$

where $x$ is generated from distribution $\rho$, $\{y^w, y^l\}$ are collected with reference policy $\mu$ and $\sigma(z) = 1/(1 + \exp(-z))$ is the Sigmoid function. After the reward modeling phase, the LLM (i.e., the

policy) is fine-tuned with the learned reward $r(x, y)$, which aims to maximize the expected reward with KL-regularization:

$$\pi \leftarrow \arg\max_{\pi \in \Pi} \mathbb{E}_{x \sim \rho, y \sim \pi(\cdot|x)}[r(x, y)] - \beta \mathbb{E}_{x \sim \rho}\big[\text{KL}\big(\pi(\cdot|x)\|\pi_{\text{ref}}(\cdot|x)\big)\big], \quad (2.3)$$

where $\Pi$ denotes the policy class, $\rho$ is the distribution of prompts, and the KL regularization with parameter $\beta > 0$ is used to control the deviation of the learned policy $\pi$ from the reference policy $\pi_{\text{ref}}$. In detail, the optimization problem is usually solved with the PPO method (Schulman et al., 2017).

Later, Rafailov et al. (2023) identified the following closed-form solution to the optimization problem in (2.3):

$$\pi(\cdot|x) = \frac{1}{Z(x)} \cdot \pi_{\text{ref}}(\cdot|x) \cdot \exp\left(\frac{r(x, \cdot)}{\beta}\right),$$

where $Z(x) = \sum_y \pi_{\text{ref}}(x) \cdot \exp\big(r(x, y)/\beta\big)$ is the partition function. By reparameterizing the reward function by the policy and substituting it into the negative log-likelihood in (2.2), Rafailov et al. (2023) proposed the Direct Preference Optimization (DPO) method as follows:

$$\pi \leftarrow \arg\max_{\pi \in \Pi} \mathbb{E}_{(x, y^w, y^l) \sim \mathcal{D}}\left[\log \sigma\left(\beta \log \frac{\pi(y^w|x)}{\pi_{\text{ref}}(y^w|x)} - \beta \log \frac{\pi(y^l|x)}{\pi_{\text{ref}}(y^l|x)}\right)\right],$$

which avoids the explicit learning of a reward model. Here, for a finite dataset $\mathcal{D} = \{(x_i, y_i^w, y_i^l)_{i=1}^N\}$ and a function $f : \mathcal{X} \times \mathcal{Y} \times \mathcal{Y} \to \mathbb{R}$, we denote the empirical expectation of function $f$ with respect to the dataset $\mathcal{D}$ by $\mathbb{E}_{(x, y^w, y^l) \sim \mathcal{D}}\big[f(x, y^w, y^l)\big] = \sum_{i=1}^N f(x_i, y_i^w, y_i^l)/N$.

# 3 ACCELERATED PREFERENCE OPTIMIZATION

In this section, we present a general framework for language model alignment, namely accelerated preference optimization (APO), which is built upon the iterative preference optimization framework.

## 3.1 ITERATIVE PREFERENCE OPTIMIZATION FRAMEWORK

Under the iterative Preference Optimization framework (Xu et al., 2023; Yuan et al., 2024b; Chen et al., 2024; Wu et al., 2024), the algorithm progressively updates the policy $\pi_t$, aiming to converge to the optimal policy. In detail, for each iteration $t \in [T]$, it designates the reference policy as the policy generated from the previous iteration, denoted by $\pi_t$. It estimates the reward model by minimizing the expected loss function $l(r, x, y^w, y^l)$ over the dataset $\mathcal{D}_t$:

$$r_t(\cdot, \cdot) \leftarrow \arg\max_{r(\cdot, \cdot)} \mathbb{E}_{(x, y^w, y^l) \sim \mathcal{D}_t}\big[\ell(r, x, y^w, y^l, \pi_t)\big], \quad (3.1)$$

Then, it updates the reference policy by solving the following KL-regularized optimization problem:

$$\widehat{\pi}_{t+1} \leftarrow \arg\max_{\pi \in \Pi} \mathbb{E}_{x \sim \rho, y \sim \pi(\cdot|x)}[r_t(x, y)] - \beta \mathbb{E}_{x \sim \rho}\big[\text{KL}\big(\pi(\cdot|x)\|\pi_t(\cdot|x)\big)\big]. \quad (3.2)$$

According to Rafailov et al. (2023), for each iteration $t \in [T]$, the optimization problem (3.2) has the following closed-form solution:

$$\widehat{\pi}_{t+1}(y|x) \propto \pi_t(y|x) \cdot \exp\left(\frac{r_t(x, y)}{\beta}\right).$$

Thus, we can reparamterize the reward function for each policy $\pi$ as follows:

$$r_\pi(x, y) = \beta \log \frac{\pi(y|x)}{\pi_t(y|x)}.$$

With this reparameterized reward function, the previous two-step optimization process in (3.1) and (3.2) can be integrated into the following one-step direct preference optimization:

$$\widehat{\pi}_{t+1} \leftarrow \arg\min_{r_\pi \in \mathcal{R}_t} \mathbb{E}_{(x, y^w, y^l) \sim \mathcal{D}_t}\big[\ell(r_\pi, x, y^w, y^l, \pi_t)\big], \quad (3.3)$$

where $\mathcal{D}_t$ represents the data collected at iteration $t$ using the reference policy $\pi_t$.

For the vanilla iterative preference optimization framework, the updated policy $\widehat{\pi}_{t+1}$ is directly used as the reference policy in the next iteration, where $\pi_{t+1} = \widehat{\pi}_{t+1}$. In this situation, the iterative

---

**Algorithm 1** APO (Accelerated Preference Optimization)

---

1: **input:** Reference policy $\pi_{\text{ref}}$, learning rate $\beta$, Nesterov's extrapolation parameter $\alpha$, number of iterations $T$
2: Initialize $\pi_0 = \widehat{\pi}_0 = \pi_{\text{ref}}$
3: **for** iteration $t = 0, \ldots, T$ **do**
4:      Collect the dataset $|\mathcal{D}_t| = N$ with prompts distribution $x \sim \rho$ and current reference policy $\pi_t$

5:      Set the reparameterized reward function class as following:

$$\mathcal{R}_t \leftarrow \left\{ r_\pi(x, y) = \beta \log \frac{\pi(y|x)}{\pi_t(y|x)} \Big| \pi \in \Pi \right\}$$

6:      Update the policy $\widehat{\pi}_{t+1}$ as following:

$$\widehat{\pi}_{t+1} \leftarrow \arg\min_{\pi \in \Pi} \mathbb{E}_{(x, y^w, y^l) \sim \mathcal{D}_t} \left[ \ell(r_\pi, x, y^w, y^l, \pi_t) \right] \tag{3.5}$$

7:      Compute the policy $\pi_{t+1}$ with an extrapolation step

$$\pi_{t+1}(y|x) \propto \widehat{\pi}_{t+1}(y|x) \cdot \left( \widehat{\pi}_{t+1}(y|x) / \widehat{\pi}_t(y|x) \right)^\alpha \tag{3.6}$$

8: **end for**
9: **output:** final policy $\widehat{\pi}_{T+1}$

---

optimization of the policy resembles the (Bregman) Proximal Point Method (Bregman, 1967; Censor & Zenios, 1992; Kiwiel, 1997), which iteratively minimizes the following proximal subproblem:

$$\pi_{t+1} \leftarrow \arg\min_{\pi \in \Pi} \left\{ f_t(\pi) = L_t(\pi) + \beta D(\pi, \pi_t) \right\} \tag{3.4}$$

for a given regularization parameter $\beta$ and Bregman divergence $D(\cdot, \cdot)$. In this reduction, the expected reward $-\mathbb{E}_{x \sim \rho, y \sim \pi(\cdot|x)}[r_t(x, y)]$ corresponds to the target function $L_t(\pi)$ in the proximal function $f_t(\pi)$, and the Bregman divergence $D(\pi, \pi_t)$ is chosen as the Kullback–Leibler (KL) divergence to the behavior policy $\pi_t$: $\mathbb{E}_{x \sim \rho} \left[ \text{KL}\left( \pi(\cdot|x) | \pi_t(\cdot|x) \right) \right]$.

**The Choice of Loss Function** By choosing different loss function $\ell(r_\pi, x, y^w, y^l, \pi_t)$ in (3.3), we can recover many existing iterative preference optimization algorithms. In detail, the loss function depends on the preference model $\mathbb{P}$, and we provide several concrete examples of the loss function and the corresponding preference optimization algorithms as follows.

**Example 3.1** (DPO). If we choose the loss function in (3.3) as:

$$\ell_{\text{DPO}}(r_\pi, x, y^w, y^l, \pi_t) = -\log \sigma \left( r_\pi(x, y^w) - r_\pi(x, y^l) \right),$$

it recovers DPO (Rafailov et al., 2023).

**Example 3.2** (SPPO). If we choose the loss function in (3.3) as:

$$\ell_{\text{SPPO}}(r_\pi, x, y^w, y^l, \pi_t) = \frac{1}{2} \left( r_\pi(x, y^w) - 1 + \log Z_{\pi_t}(x) \right)^2 + \frac{1}{2} \left( r_\pi(x, y^l) + \log Z_{\pi_t}(x) \right)^2,$$

where $Z_{\pi_t}(x) = \sum_{y \in \mathcal{Y}} \pi_t(y|x) \exp \left( \eta \mathbb{P}(y \succ \pi_t|x) \right)$ represents the partition function for behavior policy $\pi_t$, it recovers the Self-Play Preference Optimization (SPPO) algorithm (Wu et al., 2024). (See Appendix B.1 for a more detailed discussion.)

**Example 3.3** (IPO). If we choose the loss function in (3.3) as

$$\ell_{\text{IPO}}(r_\pi, x, y^w, y^l, \pi_t) = \left( r_\pi(x, y^w) - r_\pi(x, y^l) - \tau^{-1} \right)^2,$$

where $\tau$ is a regularization parameter, it recovers the Identity Preference Optimization (IPO) algorithm (Azar et al., 2024).

## 3.2 Accelerated Preference Optimization

So far, we have demonstrated that the iterative preference optimization framework resembles the proximal point method. For standard optimization problems, it is well known that Nesterov's momentum method (Nesterov, 1983; 2008; 2013) can accelerate the optimization algorithm both theoretically and empirically. In particular, Lin et al. (2018) proposed a framework called Catalyst, which extends Nesterov's momentum method to the proximal point method and has shown that it can accelerate it provably. In the Catalyst method, after solving the proximal operator

$$x_{t+1} = \arg\min_x \left\{ f_t(x) = f(x) + \kappa D(x, y_t) \right\},$$

where $f(x)$ is the target function and $D(x, y_t)$ is the Bregman divergence, an extrapolation step is introduced as follows:

$$y_{t+1} = x_{t+1} + \alpha_t (x_{t+1} - x_t),$$

where $\alpha_t$ is the Nesterov's extrapolation parameter and $x_{t+1} - x_t$ denotes the momentum from the previous update. Following the above idea, we introduce an extrapolation step after solving (3.3) to obtain $\pi_{t+1}$:

$$\log \pi_{t+1}(y|x) = \log \widehat{\pi}_{t+1} + \alpha \big( \log \widehat{\pi}_{t+1} - \log \widehat{\pi}_t \big), \tag{3.7}$$

where $\alpha > 0$ is the fixed Nesterov's extrapolation parameter. After normalizing the policy $\pi_{t+1}(y|x)$, we obtain

$$\pi_{t+1}(y|x) \propto \widehat{\pi}_{t+1}(y|x) \cdot \big( \widehat{\pi}_{t+1}(y|x)/\widehat{\pi}_t(y|x) \big)^\alpha$$
$$= \frac{1}{Z'_t(x)} \cdot \widehat{\pi}_{t+1}(y|x) \cdot \big( \widehat{\pi}_{t+1}(y|x)/\widehat{\pi}_t(y|x) \big)^\alpha,$$

where $Z'_t(x) = \sum_y \widehat{\pi}_{t+1}(y|x) \cdot \big( \widehat{\pi}_{t+1}(y|x)/\widehat{\pi}_t(y|x) \big)^\alpha$ represents the partition function.

Putting together all the key components discussed above, we present the APO framework in Algorithm 1.

We notice that the extrapolation step in (3.7) is similar to the model extrapolation technique introduced by Zheng et al. (2024a), which aims to develop a stronger model by extrapolating from the aligned model and the SFT model. However, there are several notable differences between APO and ExPO. First, the extrapolation step in ExPO is based on the strong assumption that a medium-aligned model can be interpolated between a weaker model and a stronger model–an assumption that lacks theoretical support. In contrast, APO is based on the observation that iterative preference optimization resembles the (Bregman) Proximal Point method and the extrapolation step follows the Nesterov's momentum technique (Nesterov, 1983; Lin et al., 2018), which has provable guarantee. Regarding algorithm design, APO is an iterative algorithm for policy optimization. For the final iteration $T$, similar to the Catalyst algorithm, APO outputs the policy $\widehat{\pi}_{T+1}$ before the extrapolation step. In contrast, ExPO is an one-shot algorithm which only has a single iteration and outputs the final policy after an extrapolation step. In this sense, ExPO can be seen as a special case of APO with only one iteration.

## 4 Theoretical Analysis

In this section, we provide a theoretical analysis of APO in Algorithm 1.

We begin with the following theorem, which outlines the optimization dynamics of the policy $\widehat{\pi}_{t+1}$ over different iterations $t \in [T]$.

**Theorem 4.1.** Suppose that $\widehat{\pi}_{t+1}(y|x) \cdot \big( \widehat{\pi}_{t+1}(y|x)/\widehat{\pi}_t(y|x) \big)^\alpha$ belongs to the policy class $\Pi$ for each iteration $t \in [T]$. Then, the updated policy $\widehat{\pi}_t$ in Algorithm 1 satisfies

$$\widehat{\pi}_{t+1}(y|x) = \frac{1}{Z_t(x)} \cdot \pi_{\text{ref}}(y|x) \cdot \exp\left( \frac{1}{\beta} \cdot \sum_{i=0}^t \left( \frac{1}{1-\alpha} - \frac{\alpha^{t+1-i}}{1-\alpha} \right) \cdot r_i(x, y) \right),$$

where $r_t(x, y) = \beta \log \widehat{\pi}_{t+1}(y|x) - \beta \log \pi_t(y|x)$ represents the reparameterized reward at iteration $t$, and $Z_t(x) = \sum_y \pi_{\text{ref}}(y|x) \cdot \exp\big( \sum_{i=0}^t (1/(1-\alpha) - \alpha^{t+1-i}/(1-\alpha)) \cdot r_i(x, y)/\beta \big)$ is the partition function.

Theorem 4.1 illustrates how the policy $\widehat{\pi}_{t+1}$ evolves with respect to the reparameterized reward $r_t(x, y)$, which is highly dependent on the choice of the loss function $\ell$ in Algorithm 1.

For the Bradley-Terry (BT) model with the loss function $\ell_{\text{DPO}}$ in Example 3.1, Theorem 1 in Rafailov et al. (2023) demonstrates that all reward functions compatible with the BT model can be expressed by the reparameterized reward. In addition, we introduce the following two assumptions, which are required by our analysis.

**Assumption 4.2** (Realizability). For each iteration $t \in [T]$ and each policy $\pi \in \Pi$, the following updated policy belongs to the policy class $\Pi$:

$$\widehat{\pi}(\cdot|x) = \frac{1}{Z_\pi(x)} \cdot \pi(\cdot|x) \cdot \exp\left(\frac{r^*(x, \cdot)}{\beta}\right) \in \Pi,$$

where $Z_\pi(x) = \sum_y \pi(x) \cdot \exp\left(r^*(x, y)/\beta\right)$ is the partition function.

**Assumption 4.3** (Boundedness). For each iteration $t \in [T]$ and each policy $\pi, \pi_t \in \Pi$, we have

$$\beta \log \frac{\pi(y|x)}{\pi_t(y|x)} \in [-R, R]$$

for all $x \in \mathcal{X}, u \in \mathcal{Y}$.

Similar assumptions have been used in Rosset et al. (2024) to provide an analysis of the statistical error for the reparameterized reward. Equipped with these assumptions, we have the following performance guarantee for APO.

**Theorem 4.4** (APO with $\ell_{\text{DPO}}$). For the Bradley-Terry (BT) model with loss function $\ell_{\text{DPO}}$, under Assumptions 4.2 and 4.3, with probability at least $1 - \delta$, the sub-optimality gap between $\widehat{\pi}_{t+1}$ and the optimal policy $\pi^*(x) = \arg\max_{y \in \mathcal{Y}} r^*(x, y)$ is bounded by

$$\mathbb{E}_{x \sim \rho, y \sim \pi^*(\cdot|x)}\left[r^*(x, y)\right] - \mathbb{E}_{x \sim \rho, y \sim \widehat{\pi}_{T+1}(\cdot|x)}\left[r^*(x, y)\right]$$

$$\leq \widetilde{O}\left(\frac{(1-\alpha)\beta}{T}\right) + O\left(\sqrt{\frac{(T+1)\sum_{t=0}^{T} \kappa_t \cdot \log\left(T|\Pi|/\delta\right)}{N\beta^2(1-\alpha)^2}}\right),$$

where the coverage coefficient $\kappa_t$ is defined as:

$$\kappa_t = \max_{(x,y) \in \mathcal{X} \times \mathcal{Y}} \frac{\widehat{\pi}_{T+1}(y|x)\pi^*_{T+1}(y|x)}{\pi_t^2(y|x)}. \tag{4.1}$$

**Remark 4.5.** The sub-optimality gap in Theorem 4.4 consists of two terms: the optimization error $\widetilde{O}\left((1-\alpha)\beta/T\right)$ and the statistical error $\widetilde{O}\left(\sqrt{T/(N\beta^2(1-\alpha)^2)}\right)$, and there exists a tradeoff between these two errors. Specifically, a larger extrapolation parameter $\alpha \to 1$ will decrease the optimization error but increase the statistical error. Moreover, when the dataset size is sufficiently large ($N \to \infty$), the optimization error becomes the dominant term in determining the sub-optimality gap and the reduction in optimization error outweighs the increase in statistical error. Under this situation, Theorem 4.4 suggests a $\widetilde{O}\left((1-\alpha)\beta/T\right)$ sub-optimality gap for Algorithm 1 with the loss function $\ell_{\text{DPO}}$. Compared with the standard iterative DPO method with $\alpha = 0$, Algorithm 1 improves the sub-optimality gap by a factor of $1/(1-\alpha)$ thanks to the Nesterov's momentum technique.

Theorem 4.4 only provides theoretical guarantee on the sub-optimality gap for the policy $\widehat{\pi}_{T+1}$. In order to derive the convergence of policy $\widehat{\pi}_{T+1}$ to the optimal policy $\pi^*$, we need the following minimal sub-optimality gap assumption.

**Assumption 4.6** (Minimal sub-optimality gap). For each prompt $x \in \mathcal{X}$, let $y_x^* = \arg\max_{y \in \mathcal{Y}} r^*(x, y)$ be the optimal response. The sub-optimality gap between the optimal response and any other responses is strictly positive: $\min_{y \neq y_x^*} r^*(x, y_x^*) - r^*(x, y) \geq \Delta > 0$

**Remark 4.7.** For a general Bradley-Terry (BT) model without the minimal sub-optimality gap, there may exist multiple responses that share the same maximum reward, i.e., $r(x, y_1) = r(x, y_2) = \max_{y \in \mathcal{Y}} r^*(x, y)$. In this case, the optimal policy is not unique, and it is impossible to show that the policy $\widehat{\pi}_{T+1}$ converges to a specific optimal policy.

**Theorem 4.8** (APO with $\ell_{\text{DPO}}$). For the Bradley-Terry (BT) model with loss function $\ell_{\text{DPO}}$, under Assumptions 4.2, 4.3 and 4.6, with probability at least $1 - \delta$, the TV-distance between $\widehat{\pi}_{t+1}$ and the optimal policy $\pi^*(x) = \arg\max_{y \in \mathcal{Y}} r^*(x, y)$ is bounded by

$$\mathbb{E}_{x \sim \rho}\left[D_{\text{TV}}\left(\widehat{\pi}_{T+1}(\cdot|x), \pi^*(\cdot|x)\right)\right]$$

$$\leq \exp\left(-O\left(\frac{T\Delta}{(1-\alpha)}\cdot\frac{1}{\beta}\right)\right) + O\left(\sqrt{\frac{(T+1)\sum_{t=0}^{T}\kappa_t\cdot\log\left(T|\Pi|/\delta\right)}{N\beta^2(1-\alpha)^2}}\right),$$

where the coverage coefficient $\kappa_t$ is defined in (4.1).

**Remark 4.9.** The TV-distance between $\widehat{\pi}_{t+1}$ and the optimal policy $\pi^*(x) = \arg\max_{y\in\mathcal{Y}} r^*(x,y)$ includes both the optimization error $\exp\left(-O(T/(1-\alpha))\right)$ and the statistical error $\widetilde{O}\left(\sqrt{T/(N(1-\alpha)^2)}\right)$. Similar to Theorem 4.4, there is a tradeoff between these two errors based on the choice of $\alpha$. In addition, when the dataset size is sufficiently large ($N\to\infty$), Theorem 4.8 suggests that APO converges to the optimal policy at a rate of $\exp\left(-O(T/(1-\alpha))\right)$. In comparison, iterative DPO is a special case of our algorithm with $\alpha = 0$, which converges to the optimal policy at a slower rate of $\exp\left(-O(T)\right)$.

## 5 EXPERIMENTS

In this section, we detail the experimental settings used to validate the efficacy of our proposed APO algorithm. We evaluate the model's performance across various benchmark tasks and explore the impact of different momentum schedules.

### 5.1 EXPERIMENTAL SETUP

**Model and Datasets.** We use Mistral AI's `Mistral-7B-Instruct-v0.2` (Jiang et al., 2023a) as our base model, which is a fine-tuned version based on the pretrained `Mistral-7B-v0.2` (Jiang et al., 2023a) on several publicly available datasets for instruction-following. With its architecture having demonstrated robust performance improvements over `Llama2 13B Chat` (Touvron et al., 2023) in Chatbot Arena (Chiang et al., 2024), MT-Bench (Zheng et al., 2024b), and other related tasks, `Mistral-7B-Instruct-v0.2` has become a popular choice for base models in recent reinforcement learning (RL) fine-tuning research (Hoang et al., 2023; Kawin et al., 2023). For training, we employ the UltraFeedback dataset (Cui et al., 2023). Unlike traditional fine-tuning methods that depend on responses and preference labels generated by proprietary models like GPT-4, we utilize only the instruction set from UltraFeedback. All responses are autonomously generated by our model following an online principle, and the preference pairs are labeled using a separate reward model, PairRM (Jiang et al., 2023b)[1]. The instruction set used for both training and validation includes a total of 64k instructions that span a diverse range of tasks, such as UltraChat, ShareGPT, Evol-Instruct, TruthfulQA, FalseQA, and FLAN, as detailed in Cui et al. (2023). Over three training iterations, we divided the instruction set into three folds as in Hoang et al. (2023); Kawin et al. (2023), allocating approximately 20k instructions per iteration, with an additional 1k left out for validation. In each iteration, we actually allocate approximately 20k instructions for training, with an additional 1k reserved for validation. This iterative, multi-fold approach aligns with techniques used in training SPPO (Wu et al., 2024) and iterative KTO (Ethayarajh et al., 2024; Kawin et al., 2023) models, helping to improve generalization by preventing overfitting during the fine-tuning stage. Overall, our training pipeline is independent of human or GPT inputs. We defer hyperparameters to Sec. F.1.

**Evaluation.** For evaluating the performance of our model, we utilize three common benchmarking tasks: the AlpacaEval 2.0 (Li et al., 2023b), the MT-Bench (Zheng et al., 2024b), and the Open LLM Leaderboard (Beeching et al., 2023; Gao et al., 2023). Among them, AlpacaEval 2.0 is the most indicative benchmark for our method with the current choice of experimental settings, focusing on general instruction-following capabilities as assessed by `GPT-4-Turbo`, with outcomes measured by a weighted win-rate against `GPT-4-Turbo`. Another important benchmark is MT-Bench, which also targets instruction-following but offers less differentiation between models. Additionally, we present results from the Open LLM Leaderboard, which is based on accuracies of multiple-choice questions.

**AlpacaEval 2.0.** As our primary evaluation metrics, AlpacaEval 2.0 incorporates an extensive set of 805 prompts. These prompts are simplified versions derived from the AlpacaFarm (Dubois et al., 2023) evaluation set, covering a wide range of topics such as Health, Linguistics, Entertainment, Technology, Coding, Gaming, Arts, Sports, and more (Yuan et al., 2024b). During the evaluation, we consult the help of `GPT-4-Turbo` to compare the responses generated by our model with

---

[1]In our work, we only use the reward model to behave as an oracle for the preference data, and our APO algorithm operates solely on the collected preference data without accessing the underlying reward values.

Table 1: **AlpacaEval 2.0 evaluation**. Comparison of APOwith state-of-the-art iterative training algorithms. Results are reported in both raw win rate (%) and length-controlled (LC) win rate (%). Additionally, the average response character length (Avg. Len) of each model is provided. The row highlighted in light red represents the results achieved by our APO. The highest LC win rate and raw win rate, both achieved by our APO at iteration 3, are emphasized in bold. Improvements of LC win rate from the previous iteration to the current iteration are calculated and marked by a subscripted '(+)' in red.

| Model | Epochs | AlpacaEval 2.0 | | |
| --- | --- | --- | --- | --- |
| | | LC Win Rate | Win Rate | Avg. Len |
| Mistral-7B-Instruct-v0.2 | - | 17.11 | 14.72 | 1676 |
| Snorkel (Mistral-PairRM-DPO) | - | 26.39 | 30.22 | 2736 |
| Contextual AI (KTO-Mistral-PairRM) | - | 29.71 | 33.23 | 2521 |
| Self-Rewarding 70B Iter1 | - | - | 9.94 | 1092 |
| Self-Rewarding 70B Iter2 | - | - | 15.38 | 1552 |
| Self-Rewarding 70B Iter3 | - | - | 20.44 | 2552 |
| DPO Iter1 | Epoch 1 | 25.35 | 30.71 | 2369 |
| DPO Iter2 | Epoch 3 | 26.92 | 32.54 | 2529 |
| DPO Iter3 | Epoch 4 | 27.32 | 34.99 | 2847 |
| APO Iter1 ($\widehat{\pi}_1$) | Epoch 1 | 25.35$_{(+8.24)}$ | 30.71 | 2369 |
| APO Iter1 ($\pi_1$) | - | 28.23 | 35.75 | 2640 |
| APO Iter2 ($\widehat{\pi}_2$) | Epoch 1 | 29.56$_{(+4.21)}$ | 37.53 | 2636 |
| APO Iter2 ($\pi_2$) | - | 29.73 | 38.65 | 2799 |
| APO Iter3 ($\widehat{\pi}_3$) | Epoch 4 | **31.73$_{(+2.17)}$** | **39.38** | 2950 |

those produced by `GPT-4-Turbo`. The final win rate are weighted based on the uncertainty of the judge. Employing the default AlpacaEval 2.0 pipeline, this metric demonstrates a 0.93 Spearman correlation and 68.1% agreement rate with human annotators (Li et al., 2023b).

**MT-Bench**: MT-Bench comprises of 80 multi-turn questions across eight distinct dimensions: Writing, Roleplay, Reasoning, Math, Coding, Extraction, STEM, and Humanities. The evaluation process prompts the `GPT-4` judge to assign scores from 1 to 10 to the generated responses. However, this scoring metric has limitations. Particularly, score saturation occurs quickly in certain dimensions, making it difficult to distinguish between models. Additionally, this can disadvantage models that perform exceptionally well in these saturated dimensions, as their distinctiveness is less recognizable.

**Open LLM Leaderboard**: The Open LLM Leaderboard evaluates models across six different tasks, focusing on a variety of language modeling capabilities. Each task consists of multiple-choice questions (MCQs), with the most probable choice being selected and compared against the correct answer. The final score is calculated based on accuracy. The ability for answering MCQs are not directly related with instruction-following abilities. The results are deferred to Sec. F.2.

**Baselines.** Starting from `Mistral-7B-Instruct-v0.2`, we compare our method with existing iterative training results, including iterative DPO (Hoang et al., 2023) and iterative KTO (Ethayarajh et al., 2024; Kawin et al., 2023). We also present results from our own iterative DPO training. All three baselines utilize the same base model, training dataset, dataset splits, and training pipelines, which effectively highlights the differences in methodology. Additionally, as one of the pioneering approaches in iterative DPO, we include the Self-Rewarding algorithm (Yuan et al., 2024b) as another baseline. Note that different from other baselines, the training for the Self-Rewarding algorithm incorporates additional self-instruct data augmentation and is based on the LLama2 70B model, but requires no external reward model.

## 5.2 Experimental Results

Having introduced our evaluation metrics and baselines, we now turn to our training pipeline and main results. In Algorithm 1, we begin by setting $\pi_0 = \widehat{\pi}_0 = \pi_{\text{ref}}$ as the base model, `Mistral-7B-Instruct-v0.2`. At each iteration, we sample five pairs of responses under

Table 2: **MT-Bench Evaluation**. The MT-Bench comprises a total of 8 tasks. We present the average score for all 8 tasks, as well as the average score across 5 dimensions relevant to our training dataset: Writing, Roleplay, Extraction, STEM, and Humanities. Each task includes 2 progressive turns, and we provide the average score for the first turn, the second turn, and the overall average of both turns.

| Model | MT-Bench | | | Average of Five Tasks | | |
|---|---|---|---|---|---|---|
| | 1st Turn | 2nd Turn | Average | 1st Turn | 2nd Turn | Average |
| Mistral-7B-Instruct-v0.2 | **8.08** | 7.20 | **7.64** | 9.42 | 8.86 | 9.14 |
| DPO Iter1 | 7.98 | 7.29 | 7.63 | 9.56 | 9.06 | 9.31 |
| DPO Iter2 | 7.80 | 7.26 | 7.52 | 9.36 | 8.98 | 9.17 |
| DPO Iter3 | 7.61 | 7.25 | 7.43 | 9.23 | 9.04 | 9.14 |
| APO Iter1 | 7.98 | 7.29 | 7.63 | 9.56 | 9.06 | 9.31 |
| APO Iter2 | 7.95 | 7.26 | 7.60 | 9.56 | 9.23 | 9.40 |
| APO Iter3 | 7.72 | **7.33** | 7.53 | **9.69** | **9.44** | **9.57** |

the current policy $\pi_t$ and rank them using their PairRM score (Jiang et al., 2023a). We designate the top-ranked response as the winner and the bottom-ranked response as the loser. Following the proximal point update described in equation (6), we proceed with a momentum extrapolation. We note that when the policy is a softmax linear function, update (3.6) reduces to a momentum extrapolation in the parameter space. Consequently, we carry out an extrapolation in the parameter space to approximate the corresponding momentum step (3.6) in the probability space.

**AlpacaEval 2.0 Evaluation.** Table 1 summarizes the AlpacaEval 2.0 results of different models. As shown, APO surpasses all other state-of-the-art iterative training models under the same experimental setting, with a final length controlled (LC) win rate [2] of 31.73%, and a raw win rate of 39.38%. This demonstrates an increase of 1.78% in LC win rate and 1.74% in raw win rate over our implementation of iterative DPO, which only achieved an LC win rate of 29.95% and a raw win rate of 37.64%. Delving deeper into the effects of extrapolation from one iteration to the next, we observe that starting from the same iteration 1 with an LC win rate of 25.35%, APO at iteration 2 achieves a 29.56% LC win rate, which is 4.21% higher than iteration 1, and 2.64% higher than the 26.92% attained by vanilla DPO. By iteration 3, APO exibits a further 2.17% improvement over its previous iteration, maintaining a lead of 1.78% over vanilla DPO. This is significant, especially considering the increasing challenge of achieving gains at higher performance levels. Notably, the momentum acceleration step is both training-free and data-free. It maximizes the potential of a single iteration of training data, advancing further toward optimality without requiring additional inputs. While iterative training across three iterations with approximately 20k data points each often plateaus at around 30%, APO effectively surpasses this threshold by a large margin.

Next, we discuss the effect of generation length. All iterative DPO training baselines, including Snorkel's `Mistral-PairRM-DPO` and our `DPO Iter3`, exhibit a trend of increasing response length. This is inevitable because all response pairs used for training are generated online and ranked by PairRM, where longer sequences are more likely to be chosen as winners. This effect accumulates as the model trains on its own generation. We discuss this effect from the following perspectives: First, both humans and `GPT-4` exhibit length bias, and our goal is not to prevent any length growth but to evaluate it from a perspective of fairness. Therefore, we primarily refer to the LC win rate [2], which reflects the predicted win rate when response lengths are at the same level as the baseline. Second, we note that `APO Iter2` achieves a significantly higher LC win rate and overall win rate compared to Snorkel's iterative DPO implementation, while generating much shorter responses (2636 characters compared to 2736). When compared to our `DPO Iter3`, `APO Iter2` achieves a comparable win rate, but also with shorter response lengths. This verifies a faster convergence rate and enhanced performance of our methodology. Finally, this length growth is not inherent to our method but is a result of the online training and PairRM ranking mechanism. It can be mitigated if we query winner/loser pairs and preference choices from `GPT-4` or human, with much higher cost.

---

[2]The length-controlled learning rate is calculated using logistic regression, with inputs including the model weights, the instruction, and the length difference between the model and a baseline. The LC win rate is designed to correlate more strongly with human preferences and to be more robust against length attacks. Longer responses are penalized in a manner that closely simulates (by causual inference) human length preferences.

**MT-Bench Evaluation.** In addition, we also evaluate our model using the MT-Bench tasks. Among the eight tasks, three—math, coding, and reasoning—are evaluated by a `GPT-4` math judge, while the other five—writing, roleplay, extraction, STEM, and humanities—are evaluated by a `GPT-4` general judge. Table 2 illustrates the results of iterative DPO and APO across all three iterations. Since the UltraFeedback dataset is not designed for math/code tasks, and our model has never seen correct answers, the ability to solve math questions is not directly related to our training objective. Therefore, we also provide an average of scores for the five tasks focusing on general instruction-following abilities. From Table 2, we see that APO exhibits superior performance on the five general tasks, showing consistent improvement with each iteration, with average scores progressing from 9.31 to 9.40 to 9.57. In contrast, both the base model and the iteratively trained DPO model maintain an average score of around 9.14. Moreover, even when math, coding and reasoning are included, the performance of APO in the second turn surpasses that of other models. However, because it is not specifically trained on math problems, the overall average score is slightly reduced. This decline in average is primarily due to a saturation in the tasks where our method excels (achieving 9.57 out of 10), with limited room for further improvement in these areas. Consequently, the tasks where our model was not trained on become more detrimental when calculating an average of eight.

## 5.3 DISCUSSION

**Effect of Coefficient $\alpha$.** In Table 3, we explore how different choices of $\alpha$ might affect performance. At iteration 1, performance fluctuates when changing $\alpha$, peaking at $\alpha = 0.3$ with a win rate of 28.23%. Lower and higher values of $\alpha$ yield less stable results, indicating that extreme values may either underutilize or overcompensate the momentum effect. At iteration 2, the model demonstrates performance improvements at both $\alpha = 0.3$ and $\alpha = 0.6$, with the highest win rate of 30.62% observed at $\alpha = 0.6$. Although $\alpha = 0.3$ continues to provide a strong performance, it is surpassed by the higher $\alpha = 0.6$, suggesting that the optimal $\alpha$ might increase with iterations as the model stabilizes. Nevertheless, we continue to use $\alpha = 0.3$ at iteration 2, as our theoretical analysis assumes $\alpha$ as a constant. While an adaptive approach to setting $\alpha$ might offer benefits, $\alpha = 0.3$ has consistently shown to be the most stable and effective choice across different iterations.

Table 3: The effect of different extrapolation strategy and parameters at iteration 1 and 2. Results are reported in terms of length-controlled win rate (%). The row labeled "Start from $\widehat{\pi}_0$" represents results at $t = 1$ ($\pi_1$). The row labeled "Start from $\widehat{\pi}_1$" corresponds to results at $t = 2$ ($\pi_2$). Additionally, the row labeled "Start from $\pi_1$" shows results when the momentum vector is calculated using $\widehat{\pi}_2 - \pi_1$ instead of $\widehat{\pi}_2 - \widehat{\pi}_1$, which is not covered by our theoretical guarantee.

| Starting From | $\alpha =0.1$ | $\alpha =0.2$ | $\alpha =0.3$ | $\alpha =0.4$ | $\alpha =0.5$ | $\alpha =0.6$ |
|---|---|---|---|---|---|---|
| $\widehat{\pi}_0$ | 27.41 | 25.30 | 28.23 | 24.40 | 25.52 | 26.57 |
| $\widehat{\pi}_1$ | 28.11 | 29.15 | 29.73 | 28.70 | 29.33 | 30.62 |
| $\pi_1$ | 28.23 | 28.16 | 27.21 | 28.44 | 29.75 | 28.90 |

**Effect of Extrapolation Strategy.** When the momentum vector is calculated using $\widehat{\pi}_2 - \pi_1$ instead of $\widehat{\pi}_2 - \widehat{\pi}_1$, the performance appears less stable, with the lowest LC win rate being 27.21. Despite achieving competitive win rates at certain $\alpha$ values, the results are generally lower than those using the standard Nesterov's momentum scheme. Intuitively, $\widehat{\pi}_2 - \widehat{\pi}_1 = (\widehat{\pi}_2 - \pi_1) + (\pi_1 - \widehat{\pi}_1)$ includes accumulated information, acting as an exponential average of all previous updates. In contrast, if extrapolation is based on $\pi_1$, the direction's variance is significantly higher, leading to less predictable outcomes. This instability underscores the importance of adhering to the theoretically guaranteed strategy for consistent performance gains.

## 6 CONCLUSIONS AND FUTURE WORK

In this work, we studied the iterative preference optimization framework for aligning large language models (LLMs) with human preferences and showed that it resembles the proximal point method. Based on this observation, we introduced a general framework, APO, incorporating Nesterov's momentum technique. Theoretically, we show that our method achieves a faster convergence rate than the standard iterative DPO and SPPO methods. Our experimental results demonstrate the superiority of APO over iterative DPO on the AlpacaEval 2.0 benchmark and on the instruction-following tasks of MT-Bench, achieving both accelerated convergence rate, and better final performance.

**Limitation.** Due to limited computational resources, we do not evaluate APO with the SPPO (Wu et al., 2024) loss function in the current experiments, and we plan to investigate it in our future work. Additionally, while our model demonstrates consistent improvements on instruction-following tasks, it faces challenges in solving math problems. This limitation is largely due to the choice of dataset and the restriction of not utilizing additional information from GPT-4 or human sources. In the future, we aim to address this by incorporating larger datasets and leveraging GPT-4 supervision.

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

## A    RELATED WORK

### A.1    REINFORCEMENT LEARNING FROM HUMAN FEEDBACK

Learning from human feedback in reinforcement learning can be traced back to Knox & Stone (2009); Wirth et al. (2017) and was later popularized by Christiano et al. (2017), which incorporated human preferences into deep reinforcement learning. Recently, RLHF has gained popularity in natural language processing and has become a paradigmatic method for fine-tuning large language models (LLMs) (Achiam et al., 2023; Touvron et al., 2023; OpenAI, 2023) to align with human objectives. The standard process for alignment with human feedback (Ouyang et al., 2022) requires both reward modeling and policy optimization. Recently, Rafailov et al. (2023) proposed the Direct Preference Optimization method, which replaces the reward modeling process with a reparame-terized reward and directly performs policy optimization using preference data. Similarly, Kawin

et al. (2023) proposed the Kahneman-Tversky Optimization method (KTO) with a reparameterized reward, while considering different human-aware loss functions.

More recently, iterative variants of policy optimization have garnered increasing attention. Xu et al. (2023) investigated the iterative preference optimization method and proposed the PAIRWISE CRINGE algorithm, which iteratively generates a new preference dataset using the current model and then updates the model with a combination of the original preference data and the newly labeled data. Additionally, they studied the iterative version of Direct Preference Optimization (DPO) and demonstrated that it achieves better performance than standard PPO or DPO. Later, Yuan et al. (2024b) studied Self-Rewarding Language Models, where the algorithm iteratively updates the reference policy in the DPO method and uses LLM-as-a-Judge prompting (Zheng et al., 2024b) to provide its own rewards for the generated new preference data. Despite this work demonstrating the superiority of iterative optimization process, there is a lack of theoretical foundations for these practical frameworks. Recently, Xiong et al. (2023) proposed Gibbs Sampling from Human Feedback, offering theoretical analysis with the aid of linear function approximation and the incorporation of an uncertainty bonus. Compared with previous methods, the reference policy remains fixed, the newly trained model is only used to generate new preference data, and this trained model will not be inherited in the subsequent iteration. On the contrary, our research focuses on studying the iterative DPO model without incorporating an uncertainty bonus, and it continually updates the reference policy throughout the process.

Most of the works motioned above rely on the assumption that the latent preference distribution $P(y_1 \succ y_2|x)$ follows the Bradley-Terry (BT) model (Bradley & Terry, 1952)), and there exists a series of works focusing on general preference, where human preference may not strictly be transitive. Under the general preference assumption, iterative optimization (Munos et al., 2023; Swamy et al., 2024; Rosset et al., 2024; Wu et al., 2024) is also employed to find the Nash-equilibrium policy. We leave the extension of our algorithm to general preference model in the future work.

## A.2 ACCELERATED OPTIMIZATION WITH MOMENTUM

The idea of accelerating gradient methods has been extensively explored over the decades. One of the earliest contributions to this field was the Polyak momentum (Polyak, 1964), which achieved faster convergence by leveraging the history of previous iterates. However, this early approach sometimes failed to converge even for strongly convex objectives (Lessard et al., 2016). This was further refined by Nesterov's accelerated gradient (NAG) (Nesterov, 1983), with a guarantee for faster convergence rates for general smooth convex functions. Following these foundational works, other acceleration algorithms and analyses emerged for gradient descent and its proximal variants (Beck & Teboulle, 2009; Tseng, 2009; Taylor et al., 2017; Allen-Zhu & Orecchia, 2014; Bubeck et al., 2015a; Diakonikolas & Orecchia, 2017; Cohen et al., 2020). In addition, there has been another line of work in understanding Nesterov's acceleration through continuous analysis and Lyapunov function analysis (Su et al., 2016; Shi et al., 2019; Shi, 2021; Shi et al., 2022; Chen et al., 2022). Momentum techniques have also proven effective in accelerating minimax game settings (Jin et al., 2022; Kovalev et al., 2022; Thekumparampil et al., 2022; Li et al., 2023a; Yuan et al., 2024a). However, the application of acceleration techniques to LLMs remains relatively unexplored.

The Proximal Point Method (PPM) (Moreau, 1965; Martinet, 1970; Rockafellar, 1976; Güler, 1992; Bauschke & Combettes) is a fundamental methodology that achieves optimality by iteratively solving auxiliary problems in a regularized form. Generally, these auxiliary problems cannot be solved directly, and different approximation methods yield various types of acceleration (He & Yuan, 2012; Salzo et al., 2012). One of the connections between PPM and NAG is established in Ahn & Sra (2022), where the authors interpret the updates of NAG as different approximations of the proximal method. The Catalyst approach (Lin et al., 2015; 2018) represents another variant of PPM that achieves accelerated convergence through practical inexact schemes (Lin et al., 2015; Salzo et al., 2012). This method effectively integrates Nesterov's acceleration with the proximal point framework, making it applicable to a wide range of machine learning algorithms, including gradient descent and block coordinate descent. It can be applied to variance reduction algorithms (Lin et al., 2015; 2018), demonstrating its strong adaptability to various machine learning scenarios. Despite the widespread adoption of momentum acceleration in both theoretical and practical applications, we are the first to establish a connection between DPO and the proximal point method, linking iterative DPO with the Catalyst approach, and providing a comprehensive theoretical analysis of this interpretation.

## B  ACCELERATED PREFERENCE OPTIMIZATION WITH GENERAL PREFERENCES

In the previous sections, we assumed the existence of a latent reward function and focused on the Bradley-Terry (BT) model. However, (Tversky, 1969) observed that the preferences across all possible responses may not be monotonous or transitive, which cannot be represented by a latent reward function. In this section, we extend the analysis to environments with general preferences and focus on the win probability between different responses.

In detail, for an environment with general preferences, we denote the probability that response $y_1$ is preferred over response $y_2$ given the context $x$ as $\mathbb{P}(y_1 \succ y_2|x)$, and we assume that the preference model is antisymmetric, such that $\mathbb{P}(y_2 \succ y_1|x) = 1 - \mathbb{P}(y_1 \succ y_2|x)$. Under this assumption, we define the reward $r_{\pi,\pi'} = \mathbb{E}_{y_1 \sim \pi, y_2 \sim \pi'}\big[\mathbb{P}(y_1 \succ y_2|x)\big]$ as the win probability for policy $\pi$ against policy $\pi'$. Our goal is to identify the **von Neumann winner** (Dudík et al., 2015) with the preference model. Specifically, the von Neumann winner corresponds to the Nash equilibrium of the following two-player zero-sum game:

$$(\pi^*, \pi^*) = \arg\max_{\pi} \min_{\pi'} \mathbb{E}_{y_1 \sim \pi, y_2 \sim \pi'}\big[\mathbb{P}(y_1 \succ y_2|x)\big]$$

$$= \arg\max_{\pi} \min_{\pi'} r_{\pi,\pi'}.$$

### B.1  REDUCTION TO SPPO WITH $\ell_{\text{SPPO}}$

For the Self-Play Preference Optimization (SPPO) (Wu et al., 2024) algorithm, if we set the learning rate $\beta$ in Algorithm 1 to be the inverse of the learning rate $\eta$ in the SPPO algorithm, the preference optimization process can be expressed as follows:

$$\pi_{t+1} = \arg\min_{\pi} \mathbb{E}_{x \sim \rho, y \sim \pi_t(\cdot|x)} \left[ \log\left(\frac{\pi(y|x)}{\pi_t(y|x)}\right) - \eta\left(\mathbb{P}(y \succ \pi_t|x) - \log Z_{\pi_t}(x)\right) \right]^2$$

$$= \arg\min_{\pi} \mathbb{E}_{x \sim \rho, y \sim \pi_t(\cdot|x)} \left[ \beta \log\left(\frac{\pi(y|x)}{\pi_t(y|x)}\right) - \left(\mathbb{P}(y \succ \pi_t|x) - \log Z_{\pi_t}(x)\right) \right]^2$$

$$= \arg\min_{r_\pi \in \mathcal{R}_t} \mathbb{E}_{x \sim \rho, y \sim \pi_t(\cdot|x)} \left[ r_\pi(x, y) - \mathbb{E}_{y' \sim \pi_t}\big[\mathbb{P}(y \succ y'|x)\big] + \log Z_{\pi_t}(x) \right]^2$$

$$= \arg\min_{r_\pi \in \mathcal{R}_t} \mathbb{E}_{x \sim \rho, y, y' \sim \pi_t(\cdot|x)} \big[ r_\pi(x, y) - \mathbb{1}(y \succ y'|x) + \log Z_{\pi_t}(x) \big]^2 + C_{\pi_t}$$

$$= \arg\min_{r_\pi \in \mathcal{R}_t} \mathbb{E}_{x \sim \rho, y, y' \sim \pi_t(\cdot|x)} \big[ r_\pi(x, y) - \mathbb{1}(y \succ y'|x) + \log Z_{\pi_t}(x) \big]^2/2$$

$$+ \mathbb{E}_{x \sim \rho, y, y' \sim \pi_t(\cdot|x)} \big[ r_\pi(x, y') - \mathbb{1}(y' \succ y|x) + \log Z_{\pi_t}(x) \big]^2/2 + C_{\pi_t},$$

where $Z_{\pi_t}(x) = \sum_{y \in \mathcal{Y}} \pi_t(y|x) \exp\big(\eta \mathbb{P}(y \succ \pi_t|x)\big)$ represents the partition function for behavior policy $\pi_t$, $C_{\pi_t} = \mathbb{E}_{x \sim \rho, y, y' \sim \pi_t(\cdot|x)}\big[\mathbb{P}(y \succ y'|x) - \mathbb{1}(y \succ y'|x)\big]^2$ is the variance of behavior policy $\pi_t$, the second equation holds due to $\beta = \eta^{-1}$ and the last equation holds because $y, y'$ collected under the same behavior policy. Therefore, the preference optimization process in SPPO is aligned with our Algorithm 1 using the SPPO loss function:

$$\ell_{\text{SPPO}}(r_\pi, x, y^w, y^l, \pi_t) = \frac{1}{2}\big(r_\pi(x, y^w) - 1 + \log Z_{\pi_t}(x)\big)^2 + \frac{1}{2}\big(r_\pi(x, y^l) + \log Z_{\pi_t}(x)\big)^2.$$

### B.2  THEORETICAL ANALYSIS WITH GENERAL PREFERENCES

In section 4, Theorems 4.4 and 4.8 analyze the performance of Algorithm 1 under the Bradley-Terry (BT) model. For a general preference model, Theorem B.2 provides a convergence rate for Algorithm 1 with the loss function $\ell_{\text{SPPO}}$ in Example 3.2, under the assumption of a minimal sub-optimality gap.

**Assumption B.1** (Minimal sub-optimality gap with general preferences)**.** For each prompt $x \in \mathcal{X}$, we assume there exist a unique optimal response $y_x^* \in \mathcal{Y}$ such that for any other sub-optimal responses $y \neq y_x^*$, we have

$$\mathbb{P}(y_x^* \succ y'|x) - \mathbb{P}(y \succ y'|x) \geq \Delta, \forall y' \in \mathcal{Y}.$$

**Theorem B.2** (APO with $\ell_{\text{SPPO}}$)**.** For general preference model with loss function $\ell_{\text{SPPO}}$, under the mild assumptions of realizability (Assumption E.3), boundedness (Assumption E.4) and Assumption B.1, with probability at least $1-\delta$, the TV-distance between $\widehat{\pi}_{t+1}$ and the optimal policy $\pi^*(x) = y_x^*$ is bounded by

$$\mathbb{E}_{x\sim\rho}\Big[\mathrm{D}_{\text{TV}}\big(\widehat{\pi}_{T+1}(\cdot|x), \pi^*(\cdot|x)\big)\Big]$$

$$\leq O\left(\sqrt{\frac{(T+1)\sum_{t=0}^{T}\kappa_t \cdot \log\big(T|\Pi|/\delta\big)}{N\beta^2(1-\alpha)^2}}\right) + \exp\left(-O\left(\frac{t\Delta}{(1-\alpha)}\cdot\frac{1}{\beta}\right)\right),$$

where the coverage coefficient $\kappa_t$ is defined as:

$$\kappa_t = \max_{(x,y)\in\mathcal{X}\times\mathcal{Y}}\frac{\widehat{\pi}_{T+1}(y|x)\pi_{T+1}^*(y|x)}{\pi_t^2(y|x)}.$$

**Remark B.3.** When the dataset size is sufficiently large ($N \to \infty$), Theorem B.2 suggests that APO converges to the optimal policy at a rate of $\exp\big(-O(T/(1-\alpha))\big)$. When $\alpha = 0$, our algorithm reduces to the standard SPPO algorithm in Wu et al. (2024), which only converges to the optimal policy at a slower rate of $\exp\big(-O(T)\big)$. It is worth noting that Wu et al. (2024) provides a $\widetilde{O}(1/\sqrt{T})$ sub-optimality gap guarantee for the average policy $\bar{\pi} = \frac{1}{T}\sum_{t=1}^{T}\pi_t$. In comparison, our result relies on a minimal sub-optimality gap assumption (Assumption B.1) and provides a faster convergence rate between the final policy $\widehat{\pi}_{t+1}$ and the optimal policy $\pi^*$. As discussed in Remark 4.7, this assumption is used to ensure the uniqueness of the optimal policy, which is necessary for the convergence guarantee.

# C  PROOF OF MAIN RESULTS

## C.1  PROOF OF THEOREM 4.1

In this section, we provide the proof of Theorem 4.1, which is crucial for understanding the optimization dynamics of Algorithm 1.

*Proof of Theorem 4.1.* Based on the definition of the reparameterized reward $r_t$, we have

$$r_t(x,y) = \beta \log \widehat{\pi}_{t+1}(y|x) - \beta \log \pi_t(y|x). \tag{C.1}$$

Furthermore, the extrapolation step (3.6) satisfies

$$\pi_{t+1}(y|x) = \frac{1}{Z_t'(x)}\cdot\widehat{\pi}_{t+1}(y|x)\cdot\big(\widehat{\pi}_{t+1}(y|x)/\widehat{\pi}_t(y|x)\big)^{\alpha},$$

where $Z_t'(x) = \sum_y \widehat{\pi}_{t+1}(y|x)\cdot\big(\widehat{\pi}_{t+1}(y|x)/\widehat{\pi}_t(y|x)\big)^{\alpha}$ represents the partition function. Taking the logarithm of both sides yields the following equation

$$\log \pi_{t+1}(y|x) = (1+\alpha)\log\widehat{\pi}_{t+1}(y|x) - \alpha\log\widehat{\pi}_t(y|x) - \log Z_t'(x). \tag{C.2}$$

For simplicity, we define $l_t(x,y) = \beta\log\widehat{\pi}_{t+1}(y|x) - \beta\log\widehat{\pi}_t(y|x)$, and thus we have

$$\begin{aligned}
l_{t+1}(x,y) &= \beta\log\widehat{\pi}_{t+2}(y|x) - \beta\log\widehat{\pi}_{t+1}(y|x)\\
&= r_{t+1}(x,y) + \beta\log\pi_{t+1}(y|x) - \beta\log\widehat{\pi}_{t+1}(y|x)\\
&= r_{t+1}(x,y) + \alpha\big(\beta\log\widehat{\pi}_{t+1}(y|x) - \beta\log\widehat{\pi}_t(y|x)\big) - \beta\log Z_t'(x)\\
&= r_{t+1}(x,y) - \beta\log Z_t'(x) + \alpha l_t(x,y), \tag{C.3}
\end{aligned}$$

where the second equation holds due to (C.1) and the third equation holds due to (C.2). Recursively using (C.3) over all iterations, we derive the following equation:

$$\begin{aligned}
l_t(x,y) &= r_t(x,y) - \beta\log Z_{t-1}'(x) + \alpha l_{t-1}(x,y)\\
&= r_t(x,y) - \beta\log Z_{t-1}'(x) + \alpha r_{t-1}(x,y) - \alpha\beta\log Z_{t-2}'(x) + \alpha^2 l_{t-2}(x,y)\\
&= \ldots\\
&= \sum_{i=0}^{t}\alpha^{t-i}\cdot r_t(x,y) - \alpha^{t-i}\cdot\beta\log Z_{i-1}'(x), \tag{C.4}
\end{aligned}$$

where we assume $Z'_{-1}(x) = 1$ for simplicity. Finally, by summing (C.4) over all iterations $0 \leq j \leq t$, we have

$$\beta \log \widehat{\pi}_{t+1}(y|x) - \beta \log \widehat{\pi}_1(y|x)$$

$$= \sum_{j=0}^{t} \sum_{i=0}^{j} \alpha^{j-i} r_i(x,y) + \alpha^{j-i} \cdot \beta \log Z'_{i-1}(x)$$

$$= \sum_{i=0}^{t} \sum_{j=i}^{t} \alpha^{j-i} r_i(x,y) + \alpha^{j-i} \cdot \beta \log Z'_{i-1}(x)$$

$$= \sum_{i=0}^{t} \left( \frac{1}{1-\alpha} - \frac{\alpha^{t+1-i}}{1-\alpha} \right) \cdot r_i(x,y) + \left( \frac{1}{1-\alpha} - \frac{\alpha^{t+1-i}}{1-\alpha} \right) \cdot \beta \log Z'_{i-1}(x). \qquad \text{(C.5)}$$

Given that $\widehat{\pi}_1 = \pi_{\text{ref}}$, we have

$$\widehat{\pi}_{t+1}(y|x)$$

$$= \pi_{\text{ref}}(y|x) \cdot \exp \left\{ \beta \cdot \sum_{i=0}^{t} \left( \frac{1}{1-\alpha} - \frac{\alpha^{t+1-i}}{1-\alpha} \right) \cdot r_t + \left( \frac{\alpha}{1-\alpha} - \frac{\alpha^{t+2-i}}{1-\alpha} \right) \cdot \beta \log Z'_{i-1}(x) \right\}$$

$$\propto \pi_{\text{ref}}(y|x) \cdot \exp \left\{ \beta \cdot \sum_{i=0}^{t} \left( \frac{1}{1-\alpha} - \frac{\alpha^{t+1-i}}{1-\alpha} \right) \cdot r_t \right\}.$$

Thus, we complete the proof of Theorem 4.1. $\qquad \square$

## C.2 PROOF OF THEOREM 4.4

In this section, we provide the proof of Theorem 4.4. To simplify the notation, we define the auxiliary policy, which is updated following the dynamics of $\widehat{\pi}_{t+1}$ in Theorem D.1, but using the latent reward $r^*(x,y)$ instead of the reparameterized reward $r_t(x,y)$:

$$\pi^*_{t+1}(y|x) = \frac{1}{Z^*_t(x)} \cdot \pi_{\text{ref}}(y|x) \cdot \exp \left( \frac{1}{\beta} \cdot \sum_{i=0}^{t} \left( \frac{1}{1-\alpha} - \frac{\alpha^{t+1-i}}{1-\alpha} \right) \cdot r^*(x,y) \right), \qquad \text{(C.6)}$$

where $Z^*_t(x) = \sum_y \pi_{\text{ref}}(y|x) \cdot \exp \left( \sum_{i=0}^{t} (1/(1-\alpha) - \alpha^{t+1-i}/(1-\alpha)) \cdot r^*(x,y)/\beta \right)$ is the partition function for the auxiliary policy. With this notation, the following lemma provide a upper bound for the statistical errors arising from the gap between the reparameterized reward $r_t(x,y)$ and the latent reward $r^*(x,y)$.

**Lemma C.1.** Under the mild assumptions of realizability (Assumption 4.2) and boundedness (Assumption 4.3), with probability at least $1-\delta$, the TV-distance between the policy $\widehat{\pi}_{T+1}$ and auxiliary policy $\pi^*_{T+1}$ is upper bounded by:

$$\mathbb{E}_{x \sim \rho} \left[ D_{\text{TV}} \left( \widehat{\pi}_{T+1}(\cdot|x), \pi^*_{T+1}(\cdot|x) \right) \right] \leq O \left( \sqrt{\frac{(T+1) \sum_{t=0}^{T} \kappa_t \cdot \log \left( T|\Pi|/\delta \right)}{N\beta^2(1-\alpha)^2}} \right),$$

where the coverage coefficient $\kappa_t$ is defined as:

$$\kappa_t = \max_{(x,y) \in \mathcal{X} \times \mathcal{Y}} \frac{\widehat{\pi}_{T+1}(y|x) \pi^*_{T+1}(y|x)}{\pi^2_t(y|x)}.$$

With the help of Lemma C.1, we start the proof of Theorem 4.4.

*Proof of Theorem 4.4.* For each iteration $t \in [T]$ and prompt $x \in \mathcal{X}$, according to the definition of $\pi^*_{t+1}$ in (C.6), we have

$$\pi^*_{t+1}(y|x) = \frac{1}{Z^*_t(x)} \cdot \pi_{\text{ref}}(y|x) \cdot \exp \left( \frac{1}{\beta} \cdot \sum_{i=0}^{t} \left( \frac{1}{1-\alpha} - \frac{\alpha^{t+1-i}}{1-\alpha} \right) \cdot r^*(x,y) \right)$$

$$\propto \pi_{\text{ref}}(y|x) \cdot \exp\left(\frac{1}{\beta} \cdot \sum_{i=0}^{t}\left(\frac{1}{1-\alpha} - \frac{\alpha^{t+1-i}}{1-\alpha}\right) \cdot r^*(x,y)\right)$$

$$= \pi_{\text{ref}}(y|x) \cdot \exp\left(\frac{1}{\beta} \cdot \left(\frac{t+1}{1-\alpha} - \frac{\alpha}{(1-\alpha)^2} + \frac{\alpha^{t+2}}{(1-\alpha)^2}\right) \cdot r^*(x,y)\right).$$

For simplicity, we set $\gamma = ((t+1)/(1-\alpha) - \alpha/(1-\alpha)^2 + \alpha^{t+1}/(1-\alpha)^2)/\beta$, and the sub-optimality gap for the policy $\widehat{\pi}_{t+1}$ and a fixed prompt $x$ can be denoted as:

$$\mathbb{E}_{y\sim\pi^*(\cdot|x)}\left[r^*(x,y)\right] - \mathbb{E}_{y\sim\pi^*_{t+1}(\cdot|x)}\left[r^*(x,y)\right]$$

$$= r^*(x, y_x^*) - \frac{\sum_{y\in\mathcal{Y}} \pi_{\text{ref}}(y|x) \cdot \exp\left(\gamma \cdot r^*(x,y)\right) \cdot r^*(x,y)}{\sum_{y'\in\mathcal{Y}} \pi_{\text{ref}}(y'|x) \cdot \exp\left(\gamma \cdot r^*(x,y')\right)}$$

$$= \frac{\sum_{y\neq y_x^*} \pi_{\text{ref}}(y|x) \cdot \exp\left(\gamma \cdot r^*(x,y) - \gamma \cdot r^*(x, y_x^*)\right) \cdot \left(r^*(x, y_x^*) - r^*(x,y)\right)}{\sum_{y'\in\mathcal{Y}} \pi_{\text{ref}}(y'|x) \cdot \exp\left(\gamma \cdot r^*(x,y') - \gamma \cdot r^*(x, y_x^*)\right)},$$

where $y_x^* = \arg\max_{y\in\mathcal{Y}} r^*(x,y)$ denotes the optimal response given the prompt $x$. For simplicity, we set $z(x,y) = \exp\left(\gamma \cdot r^*(x,y) - \gamma \cdot r^*(x, y_x^*)\right)$ and $Z(x) = \sum_{y\neq y_x^*} \pi_{\text{ref}}(y|x) \cdot z(x,y)$. With this notation, we have

$$\mathbb{E}_{y\sim\pi^*(\cdot|x)}\left[r^*(x,y)\right] - \mathbb{E}_{y\sim\pi^*_{t+1}(\cdot|x)}\left[r^*(x,y)\right]$$

$$= -\frac{\sum_{y\neq y_x^*} \pi_{\text{ref}}(y|x) \cdot z(x,y) \cdot \log z(x,y)}{\gamma \cdot \sum_{y'\in\mathcal{Y}} \pi_{\text{ref}}(y'|x) \cdot z(x,y)}$$

$$= -\frac{\left(1 - \pi_{\text{ref}}(y_x^*|x)\right) \cdot \sum_{y\neq y_x^*} \pi_{\text{ref}}(y|x)/\left(1 - \pi_{\text{ref}}(y_x^*|x)\right) \cdot z(x,y) \cdot \log z(x,y)}{\gamma \cdot \sum_{y'\in\mathcal{Y}} \pi_{\text{ref}}(y'|x) \cdot z(x,y)}$$

$$\leq -\frac{\sum_{y\neq y_x^*} \pi_{\text{ref}}(y|x) \cdot z(x,y) \cdot \log\left(\sum_{y\neq y_x^*} \pi_{\text{ref}}(y|x) \cdot z(x,y)/\left(1 - \pi_{\text{ref}}(y_x^*|x)\right)\right)}{\gamma \cdot \sum_{y'\in\mathcal{Y}} \pi_{\text{ref}}(y'|x) \cdot z(x,y)}$$

$$= -\frac{Z(x) \cdot \log\left(Z(x)/\left(1 - \pi_{\text{ref}}(y_x^*|x)\right)\right)}{\gamma \cdot \left(Z(x) + \pi_{\text{ref}}(y_x^*|x)\right)}, \tag{C.7}$$

where the inequality holds due to $f(x) = x\log x$ is a convex function with the fact that $\sum_{y\neq y_x^*} \pi_{\text{ref}}(y|x)/\left(1 - \pi_{\text{ref}}(y_x^*|x)\right) = 1$, and the last equation holds due to $z(x, y_x^*) = 1$. Now, we consider the following auxiliary function for $0 < Z \leq 1 - \pi_{\text{ref}}(y_x^*|x)$:

$$f(Z) = -Z \cdot \log\left(Z/\left(1 - \pi_{\text{ref}}(y_x^*|x)\right)\right) - \left(Z + \pi_{\text{ref}}(y_x^*|x)\right) \cdot \log\left(\left(1 - \pi_{\text{ref}}(y_x^*|x)\right)/\pi_{\text{ref}}(y_x^*|x)\right).$$

With basic math calculation, we have

$$f'(Z) = \log\left(\frac{1 - \pi_{\text{ref}}(y_x^*|x)}{Z}\right) - \left(1 + \log\left(\left(1 - \pi_{\text{ref}}(y_x^*|x)\right)/\pi_{\text{ref}}(y_x^*|x)\right)\right)$$

$$f(Z) \leq f\left(\left(1 - \pi_{\text{ref}}(y_x^*|x)\right)/\exp\left(1 + \log\left(\left(1 - \pi_{\text{ref}}(y_x^*|x)\right)/\pi_{\text{ref}}(y_x^*|x)\right)\right)\right) \leq 0.$$

Substituting the result $f(Z) \leq 0$ into (C.7), we have

$$\mathbb{E}_{y\sim\pi^*(\cdot|x)}\left[r^*(x,y)\right] - \mathbb{E}_{y\sim\pi^*_{t+1}(\cdot|x)}\left[r^*(x,y)\right]$$

$$\leq -\frac{Z(x) \cdot \log\left(Z(x)/\left(1 - \pi_{\text{ref}}(y_x^*|x)\right)\right)}{\gamma \cdot \left(Z(x) + \pi_{\text{ref}}(y_x^*|x)\right)}$$

$$\leq \frac{\log\left(\left(1 - \pi_{\text{ref}}(y_x^*|x)\right)/\pi_{\text{ref}}(y_x^*|x)\right)}{\gamma}$$

$$= \widetilde{O}\left(\frac{(1-\alpha)\beta}{t}\right). \tag{C.8}$$

Therefore, the sub-optimality gap between $\widehat{\pi}_{t+1}$ and the optimal policy $\pi^*(x) = \arg\max_{y\in\mathcal{Y}} r^*(x,y)$ is bounded by

$$\mathbb{E}_{x\sim\rho,y\sim\pi^*(\cdot|x)}\big[r^*(x,y)\big] - \mathbb{E}_{x\sim\rho,y\sim\widehat{\pi}_{T+1}(\cdot|x)}\big[r^*(x,y)\big]$$

$$= \mathbb{E}_{x\sim\rho,y\sim\pi^*(\cdot|x)}\big[r^*(x,y)\big] - \mathbb{E}_{x\sim\rho,y\sim\pi^*_{T+1}(\cdot|x)}\big[r^*(x,y)\big]$$

$$\quad + \mathbb{E}_{x\sim\rho,y\sim\pi^*_{T+1}(\cdot|x)}\big[r^*(x,y)\big] - \mathbb{E}_{x\sim\rho,y\sim\widehat{\pi}_{T+1}(\cdot|x)}\big[r^*(x,y)\big]$$

$$\leq \mathbb{E}_{x\sim\rho,y\sim\pi^*(\cdot|x)}\big[r^*(x,y)\big] - \mathbb{E}_{x\sim\rho,y\sim\pi^*_{T+1}(\cdot|x)}\big[r^*(x,y)\big]$$

$$\quad + \mathbb{E}_{x\sim\rho}\Big[2\mathrm{D}_{\mathrm{TV}}\big(\widehat{\pi}_{T+1}(\cdot|x),\pi^*_{T+1}(\cdot|x)\big)\Big]$$

$$\leq \mathbb{E}_{x\sim\rho,y\sim\pi^*(\cdot|x)}\big[r^*(x,y)\big] - \mathbb{E}_{x\sim\rho,y\sim\pi^*_{T+1}(\cdot|x)}\big[r^*(x,y)\big] + O\left(\sqrt{\frac{(T+1)\sum_{t=0}^{T}\kappa_t\cdot\log\big(T|\Pi|/\delta\big)}{N\beta^2(1-\alpha)^2}}\right)$$

$$\leq \widetilde{O}\left(\frac{(1-\alpha)\beta}{T}\right) + O\left(\sqrt{\frac{(T+1)\sum_{t=0}^{T}\kappa_t\cdot\log\big(T|\Pi|/\delta\big)}{N\beta^2(1-\alpha)^2}}\right),$$

where the first inequality holds due to the definition of TV-distance with the fact that $r^*(x,y) \in [-1,1]$, the second inequality holds due to Lemma (C.1) and the last inequality holds due to (C.8). Thus, we complete the proof of Theorem 4.4. $\qquad\square$

## C.3   Proof of Theorem 4.8

In this section, we provide the proof of Theorem 4.8.

*Proof of Theorem 4.8.* For each iteration $t \in [T]$ and prompt $x \in \mathcal{X}$, according to the definition of $\pi^*_{t+1}$ in (C.10), we have

$$\pi^*_{t+1}(y|x) = \frac{1}{Z^*_t(x)}\cdot\pi_{\mathrm{ref}}(y|x)\cdot\exp\left(\frac{1}{\beta}\cdot\sum_{i=0}^{t}\left(\frac{1}{1-\alpha}-\frac{\alpha^{t+1-i}}{1-\alpha}\right)\cdot r^*(x,y)\right)$$

$$\propto \pi_{\mathrm{ref}}(y|x)\cdot\exp\left(\frac{1}{\beta}\cdot\sum_{i=0}^{t}\left(\frac{1}{1-\alpha}-\frac{\alpha^{t+1-i}}{1-\alpha}\right)\cdot r^*(x,y)\right)$$

$$= \pi_{\mathrm{ref}}(y|x)\cdot\exp\left(\frac{1}{\beta}\cdot\left(\frac{t+1}{1-\alpha}-\frac{\alpha}{(1-\alpha)^2}+\frac{\alpha^{t+2}}{(1-\alpha)^2}\right)\cdot r^*(x,y)\right).$$

For simplicity, we set $\gamma = \big((t+1)/(1-\alpha)-\alpha/(1-\alpha)^2+\alpha^{t+1}/(1-\alpha)^2\big)/\beta$, and for each prompt $x \in \mathcal{X}$, the KL-divergence between $\pi^*_{t+1}(\cdot|x)$ and the optimal policy $\pi^*(x) = \arg\max_{y\in\mathcal{Y}} r^*(x,y)$ can be denoted as:

$$\mathrm{KL}\big(\pi^*(x)\|\pi^*_{t+1}(\cdot|x)\big) = \log\frac{1}{\pi^*_{t+1}\big(y^*_x|x\big)}$$

$$= \log\frac{\sum_{y\in\mathcal{Y}}\pi_{\mathrm{ref}}(y|x)\cdot\exp\big(\gamma\cdot r^*(x,y)\big)}{\pi_{\mathrm{ref}}(y^*_x|x)\cdot\exp\big(\gamma\cdot r^*(x,y^*_x)\big)}$$

$$\leq \frac{\sum_{y\neq y^*_x}\pi_{\mathrm{ref}}(y|x)\cdot\exp\big(\gamma\cdot r^*(x,y)\big)}{\pi_{\mathrm{ref}}(y^*_x|x)\cdot\exp\big(\gamma\cdot r^*(x,y^*_x)\big)}$$

$$= \frac{\sum_{y\neq y^*_x}\pi_{\mathrm{ref}}(y|x)\cdot\exp\big(\gamma\cdot\big(r^*(x,y)-r^*(x,y^*_x)\big)\big)}{\pi_{\mathrm{ref}}(y^*_x|x)}$$

$$\leq \frac{\max_{y\neq y^*_x}\exp\big(-\gamma\cdot\big(r^*(x,y^*_x)-r^*(x,y)\big)\big)}{\pi_{\mathrm{ref}}(y^*_x|x)}$$

$$\leq \frac{\exp(-\gamma\Delta)}{\pi_{\mathrm{ref}}(y^*_x|x)}$$

$$= \exp\left(-O\left(\frac{t\Delta}{(1-\alpha)}\cdot\frac{1}{\beta}\right)\right), \tag{C.9}$$

where $y_x^* = \arg\max_{y \in \mathcal{Y}} r^*(x, y)$ denotes the optimal response given the prompt $x$, the first inequality holds due to the fact that $\log(1+x) \leq x$, the second inequality holds due to $\sum_{y \neq y_x^*} \pi_{\text{ref}}(y|x) \leq 1$ and the last inequality holds due to Assumption 4.6. Therefore, the TV distance between optimal policy $\pi^*$ and $\widehat{\pi}_{T+1}$ can be upper bounded as following:

$$\mathbb{E}_{x \sim \rho}\Big[D_{\text{TV}}\big(\widehat{\pi}_{T+1}(\cdot|x), \pi^*(\cdot|x)\big)\Big]$$

$$\leq \mathbb{E}_{x \sim \rho}\Big[D_{\text{TV}}\big(\widehat{\pi}_{T+1}(\cdot|x), \pi_{T+1}^*(\cdot|x)\big)\Big] + \mathbb{E}_{x \sim \rho}\Big[D_{\text{TV}}\big(\pi^*(\cdot|x), \pi_{T+1}^*(\cdot|x)\big)\Big]$$

$$\leq O\left(\sqrt{\frac{(T+1)\sum_{t=0}^{T} \kappa_t \cdot \log\big(T|\Pi|/\delta\big)}{N\beta^2(1-\alpha)^2}}\right) + \mathbb{E}_{x \sim \rho}\Big[D_{\text{TV}}\big(\pi^*(\cdot|x), \pi_{T+1}^*(\cdot|x)\big)\Big]$$

$$\leq O\left(\sqrt{\frac{(T+1)\sum_{t=0}^{T} \kappa_t \cdot \log\big(T|\Pi|/\delta\big)}{N\beta^2(1-\alpha)^2}}\right) + \mathbb{E}_{x \sim \rho}\left[\sqrt{\frac{1}{2} \cdot D_{\text{KL}}\big(\pi^*(\cdot|x)\|\pi_{T+1}^*(\cdot|x)\big)}\right]$$

$$\leq O\left(\sqrt{\frac{(T+1)\sum_{t=0}^{T} \kappa_t \cdot \log\big(T|\Pi|/\delta\big)}{N\beta^2(1-\alpha)^2}}\right) + \exp\left(-O\left(\frac{t\Delta}{(1-\alpha)} \cdot \frac{1}{\beta}\right)\right),$$

where the first inequality holds due to $D_{\text{TV}}(X, Y) \leq D_{\text{TV}}(X, Z) + D_{\text{TV}}(Z, Y)$, the second inequality holds due to Lemma C.1, the third inequality holds due to $D_{\text{TV}}(X, Y) \leq \sqrt{D_{\text{KL}}(X\|Y)/2}$ and the last inequality holds due to (C.9). Thus, we complete the proof of Theorem 4.8. $\square$

### C.4 PROOF OF THEOREM B.2

In this section, we provide the proof of Theorem B.2. Similar to the proof of Theorem 4.4, we define the auxiliary policy, which is updated according to the dynamics of $\widehat{\pi}_{t+1}$ as described in Theorem D.1. However, in environments with general preferences, a latent reward $r^*(x, y)$ may not exist. Instead, we use the win probability of response $y$ against the policy $\pi_t$ to replace the reparameterized reward $r_t(x, y)$. Specifically, we set:

$$\pi_{t+1}^*(y|x) = \frac{1}{Z_t^*(x)} \cdot \pi_{\text{ref}}(y|x) \cdot \exp\left(\frac{1}{\beta} \cdot \sum_{i=0}^{t}\left(\frac{1}{1-\alpha} - \frac{\alpha^{t+1-i}}{1-\alpha}\right) \cdot r_t^*(x, y)\right), \qquad \text{(C.10)}$$

where $r_t^*(x, y) = \mathbb{E}_{y' \sim \pi_t}\big[\mathbb{P}(y \succ y'|x)\big]$ and $Z_t^*(x) = \sum_y \pi_{\text{ref}}(y|x) \cdot \exp\big(\sum_{i=0}^{t}(1/(1-\alpha) - \alpha^{t+1-i}/(1-\alpha)) \cdot r_t^*(x, y)/\beta\big)$ is the partition function for the auxiliary policy. With this notation, the following lemma provide a upper bound for the statistical errors arising from the gap between the reparameterized reward $r_t(x, y)$ and the win probability $r_t^*(x, y)$.

**Lemma C.2.** Under the mild assumptions of realizability (Assumption 4.2) and boundedness (Assumption 4.3), the TV-distance between the policy $\widehat{\pi}_{T+1}$ and auxiliary policy $\pi_{T+1}^*$ is upper bounded by:

$$\mathbb{E}_{x \sim \rho}\Big[D_{\text{TV}}\big(\widehat{\pi}_{T+1}(\cdot|x), \pi_{T+1}^*(\cdot|x)\big)\Big] \leq O\left(\sqrt{\frac{(T+1)\sum_{t=0}^{T} \kappa_t \cdot \log\big(T|\Pi|/\delta\big)}{N\beta^2(1-\alpha)^2}}\right),$$

where the coverage coefficient $\kappa_t$ is defined as:

$$\kappa_t = \max_{(x,y) \in \mathcal{X} \times \mathcal{Y}} \frac{\widehat{\pi}_{T+1}(y|x)\pi_{T+1}^*(y|x)}{\pi_t^2(y|x)}.$$

With the help of Lemma C.2, we start the proof of Theorem B.2.

*Proof of Theorem B.2.* For each iteration $t \in [T]$ and prompt $x \in \mathcal{X}$, according to the definition of $\pi_{t+1}^*$ in (C.6), we have

$$\pi_{t+1}^*(y|x) = \frac{1}{Z_t^*(x)} \cdot \pi_{\text{ref}}(y|x) \cdot \exp\left(\frac{1}{\beta} \cdot \sum_{i=0}^{t}\left(\frac{1}{1-\alpha} - \frac{\alpha^{t+1-i}}{1-\alpha}\right) \cdot r_t^*(x, y)\right)$$

$$\propto \pi_{\text{ref}}(y|x) \cdot \exp\left(\frac{1}{\beta} \cdot \sum_{i=0}^{t}\left(\frac{1}{1-\alpha} - \frac{\alpha^{t+1-i}}{1-\alpha}\right) \cdot r_t^*(x, y)\right),$$

where the win probability $r_t^*(x, y) = \mathbb{E}_{y' \sim \pi_t}\big[\mathbb{P}(y \succ y'|x)\big]$. According to the Assumption B.1, for each prompt $x \in \mathcal{X}$ and behavior policy $\pi_t$, the unique optimal response $y_x^* \in \mathcal{Y}$ satisfies

$$r_t^*(x, y_x^*) = \mathbb{E}_{y' \sim \pi_t}\big[\mathbb{P}(y_x^* \succ y'|x)\big] \geq \mathbb{E}_{y' \sim \pi_t}\big[\mathbb{P}(y \succ y'|x)\big] + \Delta = r_t^*(x, y) + \Delta, \forall y \neq y_x^*. \tag{C.11}$$

For simplicity, we set $\gamma_t = \big(1/(1-\alpha) - \alpha^{t+1-i}/(1-\alpha)\big)/\beta$, and for each prompt $x \in \mathcal{X}$, the KL-divergence between $\pi_{t+1}^*(\cdot|x)$ and the optimal policy $\pi^*(x) = y_x^*$ can be denoted as:

$$\begin{aligned}
\mathrm{KL}\big(\pi^*(x)\|\pi_{t+1}^*(\cdot|x)\big) &= \log \frac{1}{\pi_{t+1}^*(y_x^*|x)} \\
&= \log \frac{\sum_{y \in \mathcal{Y}} \pi_{\mathrm{ref}}(y|x) \cdot \exp\big(\sum_{i=0}^t \gamma_i \cdot r_i^*(x, y)\big)}{\pi_{\mathrm{ref}}(y_x^*|x) \cdot \exp\big(\sum_{i=0}^t \gamma_i \cdot r_i^*(x, y_x^*)\big)} \\
&\leq \frac{\sum_{y \neq y_x^*} \pi_{\mathrm{ref}}(y|x) \cdot \exp\big(\sum_{i=0}^t \gamma_i \cdot r_i^*(x, y)\big)}{\pi_{\mathrm{ref}}(y_x^*|x) \cdot \exp\big(\sum_{i=0}^t \gamma_i \cdot r_i^*(x, y_x^*)\big)} \\
&= \frac{\sum_{y \neq y_x^*} \pi_{\mathrm{ref}}(y|x) \cdot \exp\big(\sum_{i=0}^t \gamma_i \big(r_i^*(x, y) - r_i^*(x, y_x^*)\big)\big)}{\pi_{\mathrm{ref}}(y_x^*|x)} \\
&\leq \frac{\max_{y \neq y_x^*} \exp\big(-\sum_{i=0}^t \gamma_i \big(r_i^*(x, y_x^*) - r_i^*(x, y)\big)\big)}{\pi_{\mathrm{ref}}(y_x^*|x)} \\
&\leq \frac{\exp(-\sum_{i=0}^t \gamma_i \Delta)}{\pi_{\mathrm{ref}}(y_x^*|x)} \\
&= \exp\left(-O\left(\frac{t\Delta}{(1-\alpha)} \cdot \frac{1}{\beta}\right)\right), \tag{C.12}
\end{aligned}$$

where $y_x^*$ denotes the optimal response given the prompt $x$, the first inequality holds due to the fact that $\log(1+x) \leq x$, the second inequality holds due to $\sum_{y \neq y_x^*} \pi_{\mathrm{ref}}(y|x) \leq 1$ and the last inequality holds due to (C.11). Therefore, the TV distance between optimal policy $\pi^*$ and $\widehat{\pi}_{T+1}$ can be upper bounded as following:

$$\begin{aligned}
&\mathbb{E}_{x \sim \rho}\Big[\mathrm{D}_{\mathrm{TV}}\big(\widehat{\pi}_{T+1}(\cdot|x), \pi^*(\cdot|x)\big)\Big] \\
&\leq \mathbb{E}_{x \sim \rho}\Big[\mathrm{D}_{\mathrm{TV}}\big(\widehat{\pi}_{T+1}(\cdot|x), \pi_{T+1}^*(\cdot|x)\big)\Big] + \mathbb{E}_{x \sim \rho}\Big[\mathrm{D}_{\mathrm{TV}}\big(\pi^*(\cdot|x), \pi_{T+1}^*(\cdot|x)\big)\Big] \\
&\leq O\left(\sqrt{\frac{(T+1)\sum_{t=0}^T \kappa_t \cdot \log\big(T|\Pi|/\delta\big)}{N\beta^2(1-\alpha)^2}}\right) + \mathbb{E}_{x \sim \rho}\Big[\mathrm{D}_{\mathrm{TV}}\big(\pi^*(\cdot|x), \pi_{T+1}^*(\cdot|x)\big)\Big] \\
&\leq O\left(\sqrt{\frac{(T+1)\sum_{t=0}^T \kappa_t \cdot \log\big(T|\Pi|/\delta\big)}{N\beta^2(1-\alpha)^2}}\right) + \mathbb{E}_{x \sim \rho}\left[\sqrt{\frac{1}{2} \cdot \mathrm{D}_{\mathrm{KL}}\big(\pi^*(\cdot|x)\|\pi_{T+1}^*(\cdot|x)\big)}\right] \\
&\leq O\left(\sqrt{\frac{(T+1)\sum_{t=0}^T \kappa_t \cdot \log\big(T|\Pi|/\delta\big)}{N\beta^2(1-\alpha)^2}}\right) + \exp\left(-O\left(\frac{t\Delta}{(1-\alpha)} \cdot \frac{1}{\beta}\right)\right),
\end{aligned}$$

where the first inequality holds due to $\mathrm{D}_{\mathrm{TV}}(X, Y) \leq \mathrm{D}_{\mathrm{TV}}(X, Z) + \mathrm{D}_{\mathrm{TV}}(Z, Y)$, the second inequality holds due to Lemma C.2, the third inequality holds due to $\mathrm{D}_{\mathrm{TV}}(X, Y) \leq \sqrt{\mathrm{D}_{\mathrm{KL}}(X\|Y)/2}$ and the last inequality holds due to (C.12). Thus, we complete the proof of Theorem B.2. $\quad\square$

# D  PROOF OF LEMMAS IN APPENDIX C

## D.1  PROOF OF LEMMA C.1

In this subsection, we provide the proof of Lemma C.1 and first propose the following lemmas.

**Lemma D.1.** Under the mild assumptions of realizability and boundedness (see detailed definitions in Appendix E.1), for each iteration $t \in [T]$, with probability at least $1 - \delta$, the estimation error can

be upper bounded as follows:

$$\mathbb{E}_{x\sim\rho,(y_1,y_2)\sim\pi_t}\left[\left(r^*(x,y_1)-r^*(x,y_2)-r_t(x,y_1)+r_t(x,y_2)\right)^2\right]\leq O\left(\frac{\log\left(|\Pi|/\delta\right)}{N}\right),$$

where reparameterized reward $r_t(x,y)=\beta\log\widehat{\pi}_{t+1}(y|x)-\beta\log\pi_t(y|x)$.

**Lemma D.2.** For any $x\in\mathbb{R}^+$, we have $(1+x)\cdot|\log x|\geq|x-1|$.

With the help of Lemmas D.1 and D.2, we now begin the proof of Lemma C.1.

*Proof of Lemma C.1.* For each iteration $t\in[T]$ and prompt $x\in\mathcal{X}$, the TV-distance between $\widehat{\pi}_{t+1}(\cdot|x)$ and $\pi_{t+1}^*(\cdot|x)$ can be upper bounded as follows:

$$\mathrm{D}_{\mathrm{TV}}\big(\widehat{\pi}_{t+1}(\cdot|x),\pi_{t+1}^*(\cdot|x)\big)$$

$$=\mathbb{E}_{y\sim\pi_{t+1}^*(\cdot|x)}\left[\frac{1}{2}\cdot\left|\frac{\widehat{\pi}_{t+1}(y|x)}{\pi_{t+1}^*(y|x)}-1\right|\right]$$

$$\leq\mathbb{E}_{y\sim\pi_{t+1}^*(\cdot|x)}\left[\frac{1}{2}\cdot\left(1+\frac{\widehat{\pi}_{t+1}(y|x)}{\pi_{t+1}^*(y|x)}\right)\cdot\left|\log\frac{\widehat{\pi}_{t+1}(y|x)}{\pi_{t+1}^*(y|x)}\right|\right]$$

$$=\frac{1}{2}\cdot\mathbb{E}_{y\sim\pi_{t+1}^*(\cdot|x)}\left[\left|\log\frac{\widehat{\pi}_{t+1}(y|x)}{\pi_{t+1}^*(y|x)}\right|\right]+\frac{1}{2}\cdot\mathbb{E}_{y\sim\widehat{\pi}_{t+1}(\cdot|x)}\left[\left|\log\frac{\widehat{\pi}_{t+1}(y|x)}{\pi_{t+1}^*(y|x)}\right|\right]$$

$$\leq\sqrt{\frac{1}{2}\cdot\mathbb{E}_{y\sim\pi_{t+1}^*(\cdot|x)}\left[\left|\log\frac{\widehat{\pi}_{t+1}(y|x)}{\pi_{t+1}^*(y|x)}\right|\right]^2+\frac{1}{2}\cdot\mathbb{E}_{y\sim\widehat{\pi}_{t+1}(\cdot|x)}\left[\left|\log\frac{\widehat{\pi}_{t+1}(y|x)}{\pi_{t+1}^*(y|x)}\right|\right]^2}$$

$$\leq\sqrt{\frac{1}{2}\cdot\mathbb{E}_{y\sim\pi_{t+1}^*(\cdot|x)}\left[\left(\log\frac{\widehat{\pi}_{t+1}(y|x)}{\pi_{t+1}^*(y|x)}\right)^2\right]+\frac{1}{2}\cdot\mathbb{E}_{y\sim\widehat{\pi}_{t+1}(\cdot|x)}\left[\left(\log\frac{\widehat{\pi}_{t+1}(y|x)}{\pi_{t+1}^*(y|x)}\right)^2\right]},\quad\text{(D.1)}$$

where the first inequality holds due to Lemma D.2, the second inequality holds due to $x+y\leq\sqrt{2(x^2+y^2)}$ for $x,y>0$, and the last inequality holds due to $\mathbb{E}[x]^2\leq\mathbb{E}[x^2]$. In addition, for each iteration $t\in[T]$ and prompt $x\in\mathcal{X}$, we have

$$\mathbb{E}_{y\sim\pi_{t+1}^*(\cdot|x)}\left[\left(\log\frac{\widehat{\pi}_{t+1}(y|x)}{\pi_{t+1}^*(y|x)}\right)^2\right]+\mathbb{E}_{y\sim\widehat{\pi}_{t+1}(\cdot|x)}\left[\left(\log\frac{\widehat{\pi}_{t+1}(y|x)}{\pi_{t+1}^*(y|x)}\right)^2\right]$$

$$\leq\mathbb{E}_{y\sim\pi_{t+1}^*(\cdot|x)}\left[\left(\log\frac{\widehat{\pi}_{t+1}(y|x)}{\pi_{t+1}^*(y|x)}\right)^2\right]+\mathbb{E}_{y\sim\widehat{\pi}_{t+1}(\cdot|x)}\left[\left(\log\frac{\widehat{\pi}_{t+1}(y|x)}{\pi_{t+1}^*(y|x)}\right)^2\right]$$

$$\qquad+2\mathrm{D}_{\mathrm{KL}}\big(\widehat{\pi}_{t+1}(\cdot|x)\|\pi_{t+1}^*(\cdot|x)\big)\cdot\mathrm{D}_{\mathrm{KL}}\big(\pi_{t+1}^*(\cdot|x)\|\widehat{\pi}_{t+1}(\cdot|x)\big)$$

$$=\mathbb{E}_{y\sim\pi_{t+1}^*(\cdot|x)}\left[\left(\log\frac{\widehat{\pi}_{t+1}(y|x)}{\pi_{t+1}^*(y|x)}\right)^2\right]+\mathbb{E}_{y\sim\widehat{\pi}_{t+1}(\cdot|x)}\left[\left(\log\frac{\widehat{\pi}_{t+1}(y|x)}{\pi_{t+1}^*(y|x)}\right)^2\right]$$

$$\qquad+2\mathbb{E}_{y\sim\pi_{t+1}^*(\cdot|x)}\left[-\log\frac{\widehat{\pi}_{t+1}(y|x)}{\pi_{t+1}^*(y|x)}\right]\cdot\mathbb{E}_{y\sim\widehat{\pi}_{t+1}(\cdot|x)}\left[\log\frac{\widehat{\pi}_{t+1}(y|x)}{\pi_{t+1}^*(y|x)}\right]$$

$$=\mathbb{E}_{y\sim\pi_{t+1}^*(\cdot|x),y'\sim\widehat{\pi}_{t+1}(\cdot|x)}\left[\left(\log\frac{\widehat{\pi}_{t+1}(y|x)}{\pi_{t+1}^*(y|x)}-\log\frac{\widehat{\pi}_{t+1}(y'|x)}{\pi_{t+1}^*(y'|x)}\right)^2\right],\quad\text{(D.2)}$$

where the inequality holds due to KL-divergence is non-negative. According to Theorem 4.1 and definition of $\pi_{t+1}^*$ in (C.6), we have

$$\log\frac{\widehat{\pi}_{t+1}(y|x)}{\pi_{t+1}^*(y|x)}=\frac{\log Z_t^*(x)}{\log Z_t(x)}+\sum_{i=0}^t\frac{1}{\beta}\cdot\left(\frac{1}{1-\alpha}-\frac{\alpha^{t+1-i}}{1-\alpha}\right)\cdot\big(r_i(x,y)-r^*(x,y)\big).\quad\text{(D.3)}$$

Substituting (D.3) into (D.2), we have

$$\mathbb{E}_{y\sim\pi_{t+1}^*(\cdot|x)}\left[\left(\log\frac{\widehat{\pi}_{t+1}(y|x)}{\pi_{t+1}^*(y|x)}\right)^2\right]+\mathbb{E}_{y\sim\widehat{\pi}_{t+1}(\cdot|x)}\left[\left(\log\frac{\widehat{\pi}_{t+1}(y|x)}{\pi_{t+1}^*(y|x)}\right)^2\right]$$

$$\leq\mathbb{E}_{y\sim\pi_{t+1}^*(\cdot|x),y'\sim\widehat{\pi}_{t+1}(\cdot|x)}\left[\left(\log\frac{\widehat{\pi}_{t+1}(y|x)}{\pi_{t+1}^*(y|x)}-\log\frac{\widehat{\pi}_{t+1}(y'|x)}{\pi_{t+1}^*(y'|x)}\right)^2\right]$$

$$\leq \mathbb{E}_{y \sim \pi_{t+1}^*(\cdot|x), y' \sim \widehat{\pi}_{t+1}(\cdot|x)} \left[ \left( \sum_{i=0}^{t} \alpha_{t+1-i} \cdot \big( r_i(x, y) - r^*(x, y) \big) - \big( r_i(x, y') - r^*(x, y') \big) \right)^2 \right]$$

$$\leq \mathbb{E}_{y \sim \pi_{t+1}^*(\cdot|x), y' \sim \widehat{\pi}_{t+1}(\cdot|x)} \left[ (t+1) \cdot \sum_{i=0}^{t} \alpha_{t+1-i}^2 \cdot \Big( r_i(x, y) - r_i(x, y') - r^*(x, y) + r^*(x, y') \Big)^2 \right],$$
(D.4)

where $\alpha_i = \big( 1/(1-\alpha) - \alpha^i/(1-\alpha) \big)/\beta$ and the last inequality holds due to Cauchy–Schwarz inequality. According to the definition of coverage coefficient $\kappa_t$ in (4.1), for policy $\widehat{\pi}_{T+1}$, we have

$$\mathbb{E}_{y \sim \pi_{T+1}^*(\cdot|x)} \left[ \left( \log \frac{\widehat{\pi}_{T+1}(y|x)}{\pi_{T+1}^*(y|x)} \right)^2 \right] + \mathbb{E}_{y \sim \widehat{\pi}_{T+1}(\cdot|x)} \left[ \left( \log \frac{\widehat{\pi}_{T+1}(y|x)}{\pi_{T+1}^*(y|x)} \right)^2 \right]$$

$$\leq (T+1) \cdot$$

$$\sum_{t=0}^{T} \frac{1}{\beta^2 (1-\alpha)^2} \cdot \mathbb{E}_{y \sim \pi_{T+1}^*(\cdot|x), y' \sim \widehat{\pi}_{T+1}(\cdot|x)} \left[ \left( r_t(x, y) - r_t(x, y') - r^*(x, y) + r^*(x, y') \right)^2 \right]$$

$$\leq (T+1) \cdot$$

$$\sum_{t=0}^{T} \frac{\kappa_t}{\beta^2 (1-\alpha)^2} \cdot \mathbb{E}_{y \sim \pi_t^*(\cdot|x), y' \sim \widehat{\pi}_t(\cdot|x)} \left[ \left( r_t(x, y) - r_t(x, y') - r^*(x, y) + r^*(x, y') \right)^2 \right]$$

$$\leq \frac{1}{\sigma^2 (1+R) \cdot \big( 1 - \sigma(1+R) \big)^2} \cdot \frac{(T+1) \sum_{t=0}^{T} \kappa_t \cdot \log \big( T |\Pi|/\delta \big)}{N \beta^2 (1-\alpha)^2},$$
(D.5)

where the first inequality holds due to (D.4) with the fact that $\alpha_i \leq 1/(1-\alpha)$, the second inequality holds due to the definition of coverage coefficient $\kappa_t$ in (4.1) and the last inequality holds due to Lemma D.1 with a union bound on the probability across all $T$ iterations. Finally, substituting (D.5) into (D.1), we have

$$\mathbb{E}_{x \sim \rho} \Big[ D_{\text{TV}} \big( \widehat{\pi}_{T+1}(\cdot|x), \pi_{T+1}^*(\cdot|x) \big) \Big]$$

$$\leq \mathbb{E}_{x \sim \rho} \left[ \sqrt{\frac{1}{2} \cdot \mathbb{E}_{y \sim \pi_{T+1}^*(\cdot|x)} \left[ \left( \log \frac{\widehat{\pi}_{T+1}(y|x)}{\pi_{T+1}^*(y|x)} \right)^2 \right] + \frac{1}{2} \cdot \mathbb{E}_{y \sim \widehat{\pi}_{T+1}(\cdot|x)} \left[ \left( \log \frac{\widehat{\pi}_{T+1}(y|x)}{\pi_{T+1}^*(y|x)} \right)^2 \right]} \right]$$

$$\leq \frac{1}{\sigma(1+R) \cdot \big( 1 - \sigma(1+R) \big)} \cdot \sqrt{\frac{(T+1) \sum_{t=0}^{T} \kappa_t \cdot \log \big( T |\Pi|/\delta \big)}{N \beta^2 (1-\alpha)^2}}.$$

Thus, we complete the proof of Lemma C.1. □

### D.2 PROOF OF LEMMA C.2

In this subsection, we provide the proof of Lemma C.2 and first propose the following lemma.

**Lemma D.3.** Under the mild assumptions of realizability and boundedness (see detailed definitions in Appendix E.3), for each iteration $t \in [T]$, with probability at least $1 - \delta$, the estimation error can be upper bounded as follows:

$$\mathbb{E}_{x \sim \rho, (y_1, y_2) \sim \pi_t} \left[ \big( r_t^*(x, y_1) - r_t^*(x, y_2) - r_t(x, y_1) + r_t(x, y_2) \big)^2 \right] \leq O \left( \frac{\log \big( |\Pi|/\delta \big)}{N} \right),$$

where the win probability $r_t^*(x, y) = \mathbb{E}_{y' \sim \pi_t} \big[ \mathbb{P}(y \succ y'|x) \big]$ and reparameterized reward $r_t(x, y) = \beta \log \widehat{\pi}_{t+1}(y|x) - \beta \log \pi_t(y|x)$,

With the help of Lemma C.2, we start the proof of Lemma C.2. The proof is similar to the proof of Lemma C.1, however, it is worth noting that this proof relies on a different auxiliary policy based on the win probability rather than the latent reward:

$$\pi_{t+1}^*(y|x) = \frac{1}{Z_t^*(x)} \cdot \pi_{\text{ref}}(y|x) \cdot \exp \left( \frac{1}{\beta} \cdot \sum_{i=0}^{t} \left( \frac{1}{1-\alpha} - \frac{\alpha^{t+1-i}}{1-\alpha} \right) \cdot r_t^*(x, y) \right),$$

where the win probability $r_t^*(x,y) = \mathbb{E}_{y' \sim \pi_t}[\mathbb{P}(y \succ y'|x)]$ and $Z_t^*(x) = \sum_y \pi_{\text{ref}}(y|x) \cdot \exp\left(\sum_{i=0}^t (1/(1-\alpha) - \alpha^{t+1-i}/(1-\alpha)) \cdot r_t^*(x,y)/\beta\right)$ is the partition function for the auxiliary policy.

*Proof of Lemma C.2.* For each iteration $t \in [T]$ and prompt $x \in \mathcal{X}$, the TV-distance between $\widehat{\pi}_{t+1}(\cdot|x)$ and $\pi_{t+1}^*(\cdot|x)$ can be upper bounded as follows:

$$\mathrm{D}_{\mathrm{TV}}\big(\widehat{\pi}_{t+1}(\cdot|x), \pi_{t+1}^*(\cdot|x)\big)$$

$$= \mathbb{E}_{y \sim \pi_{t+1}^*(\cdot|x)}\left[\frac{1}{2} \cdot \left|\frac{\widehat{\pi}_{t+1}(y|x)}{\pi_{t+1}^*(y|x)} - 1\right|\right]$$

$$\leq \mathbb{E}_{y \sim \pi_{t+1}^*(\cdot|x)}\left[\frac{1}{2} \cdot \left(1 + \frac{\widehat{\pi}_{t+1}(y|x)}{\pi_{t+1}^*(y|x)}\right) \cdot \left|\log\frac{\widehat{\pi}_{t+1}(y|x)}{\pi_{t+1}^*(y|x)}\right|\right]$$

$$= \frac{1}{2} \cdot \mathbb{E}_{y \sim \pi_{t+1}^*(\cdot|x)}\left[\left|\log\frac{\widehat{\pi}_{t+1}(y|x)}{\pi_{t+1}^*(y|x)}\right|\right] + \frac{1}{2} \cdot \mathbb{E}_{y \sim \widehat{\pi}_{t+1}(\cdot|x)}\left[\left|\log\frac{\widehat{\pi}_{t+1}(y|x)}{\pi_{t+1}^*(y|x)}\right|\right]$$

$$\leq \sqrt{\frac{1}{2} \cdot \mathbb{E}_{y \sim \pi_{t+1}^*(\cdot|x)}\left[\left|\log\frac{\widehat{\pi}_{t+1}(y|x)}{\pi_{t+1}^*(y|x)}\right|\right]^2 + \frac{1}{2} \cdot \mathbb{E}_{y \sim \widehat{\pi}_{t+1}(\cdot|x)}\left[\left|\log\frac{\widehat{\pi}_{t+1}(y|x)}{\pi_{t+1}^*(y|x)}\right|\right]^2}$$

$$\leq \sqrt{\frac{1}{2} \cdot \mathbb{E}_{y \sim \pi_{t+1}^*(\cdot|x)}\left[\left(\log\frac{\widehat{\pi}_{t+1}(y|x)}{\pi_{t+1}^*(y|x)}\right)^2\right] + \frac{1}{2} \cdot \mathbb{E}_{y \sim \widehat{\pi}_{t+1}(\cdot|x)}\left[\left(\log\frac{\widehat{\pi}_{t+1}(y|x)}{\pi_{t+1}^*(y|x)}\right)^2\right]}, \quad \text{(D.6)}$$

where the first inequality holds due to Lemma D.2, the second inequality holds due to $x + y \leq \sqrt{2(x^2 + y^2)}$ for $x, y > 0$, and the last inequality holds due to $\mathbb{E}[x]^2 \leq \mathbb{E}[x^2]$. In addition, for each iteration $t \in [T]$ and prompt $x \in \mathcal{X}$, we have

$$\mathbb{E}_{y \sim \pi_{t+1}^*(\cdot|x)}\left[\left(\log\frac{\widehat{\pi}_{t+1}(y|x)}{\pi_{t+1}^*(y|x)}\right)^2\right] + \mathbb{E}_{y \sim \widehat{\pi}_{t+1}(\cdot|x)}\left[\left(\log\frac{\widehat{\pi}_{t+1}(y|x)}{\pi_{t+1}^*(y|x)}\right)^2\right]$$

$$\leq \mathbb{E}_{y \sim \pi_{t+1}^*(\cdot|x)}\left[\left(\log\frac{\widehat{\pi}_{t+1}(y|x)}{\pi_{t+1}^*(y|x)}\right)^2\right] + \mathbb{E}_{y \sim \widehat{\pi}_{t+1}(\cdot|x)}\left[\left(\log\frac{\widehat{\pi}_{t+1}(y|x)}{\pi_{t+1}^*(y|x)}\right)^2\right]$$

$$+ 2\mathrm{D}_{\mathrm{KL}}\big(\widehat{\pi}_{t+1}(\cdot|x)\|\pi_{t+1}^*(\cdot|x)\big) \cdot \mathrm{D}_{\mathrm{KL}}\big(\pi_{t+1}^*(\cdot|x)\|\widehat{\pi}_{t+1}(\cdot|x)\big)$$

$$= \mathbb{E}_{y \sim \pi_{t+1}^*(\cdot|x)}\left[\left(\log\frac{\widehat{\pi}_{t+1}(y|x)}{\pi_{t+1}^*(y|x)}\right)^2\right] + \mathbb{E}_{y \sim \widehat{\pi}_{t+1}(\cdot|x)}\left[\left(\log\frac{\widehat{\pi}_{t+1}(y|x)}{\pi_{t+1}^*(y|x)}\right)^2\right]$$

$$+ 2\mathbb{E}_{y \sim \pi_{t+1}^*(\cdot|x)}\left[-\log\frac{\widehat{\pi}_{t+1}(y|x)}{\pi_{t+1}^*(y|x)}\right] \cdot \mathbb{E}_{y \sim \widehat{\pi}_{t+1}(\cdot|x)}\left[\log\frac{\widehat{\pi}_{t+1}(y|x)}{\pi_{t+1}^*(y|x)}\right]$$

$$= \mathbb{E}_{y \sim \pi_{t+1}^*(\cdot|x), y' \sim \widehat{\pi}_{t+1}(\cdot|x)}\left[\left(\log\frac{\widehat{\pi}_{t+1}(y|x)}{\pi_{t+1}^*(y|x)} - \log\frac{\widehat{\pi}_{t+1}(y'|x)}{\pi_{t+1}^*(y'|x)}\right)^2\right], \quad \text{(D.7)}$$

where the inequality holds due to KL-divergence is non-negative. According to Theorem 4.1 and definition of $\pi_{t+1}^*$ in (C.10), we have

$$\log\frac{\widehat{\pi}_{t+1}(y|x)}{\pi_{t+1}^*(y|x)} = \frac{\log Z_t^*(x)}{\log Z_t(x)} + \sum_{i=0}^t \frac{1}{\beta} \cdot \left(\frac{1}{1-\alpha} - \frac{\alpha^{t+1-i}}{1-\alpha}\right) \cdot \big(r_i(x,y) - r^*(x,y)\big). \quad \text{(D.8)}$$

Substituting (D.8) into (D.7), we have

$$\mathbb{E}_{y \sim \pi_{t+1}^*(\cdot|x)}\left[\left(\log\frac{\widehat{\pi}_{t+1}(y|x)}{\pi_{t+1}^*(y|x)}\right)^2\right] + \mathbb{E}_{y \sim \widehat{\pi}_{t+1}(\cdot|x)}\left[\left(\log\frac{\widehat{\pi}_{t+1}(y|x)}{\pi_{t+1}^*(y|x)}\right)^2\right]$$

$$\leq \mathbb{E}_{y \sim \pi_{t+1}^*(\cdot|x), y' \sim \widehat{\pi}_{t+1}(\cdot|x)}\left[\left(\log\frac{\widehat{\pi}_{t+1}(y|x)}{\pi_{t+1}^*(y|x)} - \log\frac{\widehat{\pi}_{t+1}(y'|x)}{\pi_{t+1}^*(y'|x)}\right)^2\right]$$

$$\leq \mathbb{E}_{y \sim \pi_{t+1}^*(\cdot|x), y' \sim \widehat{\pi}_{t+1}(\cdot|x)}\left[\left(\sum_{i=0}^t \alpha^{t+1-i} \cdot \big(r_i(x,y) - r_i^*(x,y)\big) - \big(r_i(x,y') - r_i^*(x,y')\big)\right)^2\right]$$

$$\leq \mathbb{E}_{y \sim \pi_{t+1}^*(\cdot|x), y' \sim \widehat{\pi}_{t+1}(\cdot|x)}\left[(t+1) \cdot \sum_{i=0}^t \alpha^2_{t+1-i} \cdot \big(r_i(x,y) - r_i(x,y') - r_i^*(x,y) + r_i^*(x,y')\big)^2\right],$$

$$\text{(D.9)}$$

where $\alpha_i = \big(1/(1-\alpha) - \alpha^i/(1-\alpha)\big)/\beta$ and the last inequality holds due to Cauchy–Schwarz inequality. According to the definition of coverage coefficient $\kappa_t$ in (4.1), for policy $\widehat{\pi}_{T+1}$, we have

$$\mathbb{E}_{y\sim\pi^*_{T+1}(\cdot|x)}\left[\left(\log\frac{\widehat{\pi}_{T+1}(y|x)}{\pi^*_{T+1}(y|x)}\right)^2\right] + \mathbb{E}_{y\sim\widehat{\pi}_{T+1}(\cdot|x)}\left[\left(\log\frac{\widehat{\pi}_{T+1}(y|x)}{\pi^*_{T+1}(y|x)}\right)^2\right]$$

$$\leq (T+1)\cdot$$

$$\sum_{t=0}^{T}\frac{1}{\beta^2(1-\alpha)^2}\cdot\mathbb{E}_{y\sim\pi^*_{T+1}(\cdot|x),y'\sim\widehat{\pi}_{T+1}(\cdot|x)}\left[\left(r_t(x,y) - r_t(x,y') - r^*_t(x,y) + r^*_t(x,y')\right)^2\right]$$

$$\leq (T+1)\cdot$$

$$\sum_{t=0}^{T}\frac{\kappa_t}{\beta^2(1-\alpha)^2}\cdot\mathbb{E}_{y\sim\pi^*_t(\cdot|x),y'\sim\widehat{\pi}_t(\cdot|x)}\left[\left(r_t(x,y) - r_t(x,y') - r^*_t(x,y) + r^*_t(x,y')\right)^2\right]$$

$$\leq O\left(\frac{(T+1)\sum_{t=0}^{T}\kappa_t\cdot\log\big(T|\Pi|/\delta\big)}{N\beta^2(1-\alpha)^2}\right), \tag{D.10}$$

where the first inequality holds due to (D.9) with the fact that $\alpha_i \leq 1/(1-\alpha)$, the second inequality holds due to the definition of coverage coefficient $\kappa_t$ in (4.1) and the last inequality holds due to Lemma D.3 with a union bound on the probability across all $T$ iterations. Finally, substituting (D.10) into (D.6), we have

$$\mathbb{E}_{x\sim\rho}\left[D_{\mathrm{TV}}\big(\widehat{\pi}_{T+1}(\cdot|x), \pi^*_{T+1}(\cdot|x)\big)\right]$$

$$\leq \mathbb{E}_{x\sim\rho}\left[\sqrt{\frac{1}{2}\cdot\mathbb{E}_{y\sim\pi^*_{T+1}(\cdot|x)}\left[\left(\log\frac{\widehat{\pi}_{T+1}(y|x)}{\pi^*_{T+1}(y|x)}\right)^2\right] + \frac{1}{2}\cdot\mathbb{E}_{y\sim\widehat{\pi}_{T+1}(\cdot|x)}\left[\left(\log\frac{\widehat{\pi}_{T+1}(y|x)}{\pi^*_{T+1}(y|x)}\right)^2\right]}\right]$$

$$\leq O\left(\sqrt{\frac{(T+1)\sum_{t=0}^{T}\kappa_t\cdot\log\big(T|\Pi|/\delta\big)}{N\beta^2(1-\alpha)^2}}\right).$$

Thus, we complete the proof of Lemma C.2. $\qquad\square$

# E  PROOF OF LEMMAS IN APPENDIX D

## E.1  PROOF OF LEMMA D.1

In this section, we adapt the previous results on the estimation error from maximum likelihood estimation to the Bradley-Terry (BT) model (Bradley & Terry, 1952) and provide the proof of Lemma D.1.

We start the analysis with the conditional probability estimation setting. For an instance space $\mathcal{X}$ and a target space $\mathcal{Y}$, we collect the dataset $\mathcal{D} = (z_i, o_i)_{i=1}^{N}$, where $z_i$ is sampled from a reference policy $z_i \sim \mu_i$, and $o_i$ is then generated with the latent conditional density $P(o_i|z_i) = f^*(z_i, o_i)$. Under this situation, the maximum likelihood estimator across a function class $\mathcal{F} : \mathcal{Z} \times \mathcal{O} \to \mathbb{R}$ can be denoted as follows

$$f = \arg\max_{f\in\mathcal{F}}\sum_{i=1}^{N}\log f(z_i, o_i).$$

For an i.i.d. sampled dataset $\mathcal{D}$, where $\mu_1 = \mu_2 = \ldots = \mu_N$, Geer (2000) (Chapter 7) provides upper bounds for the estimation error between $f$ and the hidden function $f^*$. Later, Agarwal et al. (2020) extended these results to reinforcement learning, allowing for a martingale sampling process.

**Lemma E.1** (Theorem 21, Agarwal et al. 2020). For a finite function class $|\mathcal{F}| < +\infty$, if the latent function $f^* \in \mathcal{F}$, then with probability at least $1 - \delta$, we have

$$\sum_{i=1}^{N}\mathbb{E}_{z\sim\mu_i}\big\|f(z,\cdot) - f^*(z,\cdot)\big\|_{\mathrm{TV}}^2 \leq 2\log\big(|\mathcal{F}|/\delta\big).$$

With the help of lemma E.1, we start to prove Lemma D.1.

*Proof of Lemma D.1.* Based on the Bradley-Terry (BT) model defined in equation (2.1), for the prompt $x_i \in \mathcal{X}$ and the generated responses $y_1, y_2$, we have

$$P(y_1 \succ y_2 | x) = \sigma\big(r^*(x_i, y_{i,1}) - r^*(x_i, y_{i,2})\big).$$

In this context, we define the instance $z_i$ and target $o_i$ as follows

$$z_i = (x_i, y_{i,1}, y_{i,2}),$$
$$o_i = \mathbb{1}(y_{i,1} \succ y_{i,2}) - \mathbb{1}(y_{i,2} \succ y_{i,1}).$$

For each policy $\pi \in \Pi$, we denote the reparameterized reward $r_\pi(x, y) = \beta \log \pi(y|x) - \beta \log \pi_t(y|x)$. Thus, the density function for an instance $z = (x, y_1, y_2)$ with reward $r_\pi$ can be represented as

$$f_{r_\pi}(z, o) = \sigma\Big(o \cdot \big(r_\pi(x, y_1) - r_\pi(x, y_2)\big)\Big),$$

and we denote the density function class $\mathcal{F}$ as $\mathcal{F} = \{f_{r_\pi} | \pi \in \Pi\}$. Based on Assumption 4.2 and the definition of the Bradley-Terry (BT) model, we have $f_{r^*} \in \mathcal{F}$, and the latent density function satisfies $P(o|z) = f_{r^*}(z, o)$. Consequently, the maximum likelihood estimator can be expressed as

$$f_{r_t} = \arg \max_{f_{r_\pi} \in \mathcal{F}} \sum_{i=1}^{N} \log f_{r_\pi}(z_i, o_i),$$

where $r_t(x, y) = \beta \log \widehat{\pi}_{t+1}(y|x) - \beta \log \pi_t(y|x)$. Thus, according to Lemma E.1, the estimation error between $r^*$ and $r_t$ can be upper bounded by

$$\sum_{i=1}^{N} \mathbb{E}_{z \sim \mu_i} \big\| f_{r_t}(z, \cdot) - f_{r^*}(z, \cdot) \big\|_{\text{TV}}^2 \leq 2 \log\big(|\Pi|/\delta\big). \tag{E.1}$$

Since the dataset $\mathcal{D}_t$ is collected with reference policy $\pi_t$, we have

$$\mathbb{E}_{x \sim \rho, (y_1, y_2) \sim \pi_t} \Big[\sigma\big(r^*(x, y_1) - r^*(x, y_2)\big) - \sigma\big(r_t(x, y_1) - r_t(x, y_2)\big)\Big]^2$$
$$= \frac{1}{2} \cdot \mathbb{E}_{x \sim \rho, (y_1, y_2) \sim \pi_t} \Big[\sigma\big(r^*(x, y_1) - r^*(x, y_2)\big) - \sigma\big(r_t(x, y_1) - r_t(x, y_2)\big)\Big]^2$$
$$+ \frac{1}{2} \cdot \mathbb{E}_{x \sim \rho, (y_1, y_2) \sim \pi_t} \Big[\sigma\big(r^*(x, y_2) - r^*(x, y_1)\big) - \sigma\big(r_t(x, y_2) - r_t(x, y_1)\big)\Big]^2$$
$$= \mathbb{E}_{x \sim \rho, (y_1, y_2) \sim \pi_t} \big\| f_{r_t}(z, \cdot) - f_{r^*}(z, \cdot) \big\|_{\text{TV}}^2$$
$$\leq \frac{\log\big(|\Pi|/\delta\big)}{N}, \tag{E.2}$$

where the second equation holds due to the definition of TV-distance with $z = (x, y_1, y_2)$ and the inequality holds due to (E.1). According to Assumption 4.3, we have $r^*(x, y) \in [-1, 1]$ and $r_t(x, y) \in [-R, R]$, which implies that

$$\mathbb{E}_{x \sim \rho, (y_1, y_2) \sim \pi_t} \Big[\big(r^*(x, y_1) - r^*(x, y_2) - r_t(x, y_1) + r_t(x, y_2)\big)^2\Big]$$
$$\leq \frac{1}{\sigma^2(1+R) \cdot \big(1 - \sigma(1+R)\big)^2}$$
$$\times \mathbb{E}_{x \sim \rho, (y_1, y_2) \sim \pi_t} \Big[\sigma\big(r^*(x, y_1) - r^*(x, y_2)\big) - \sigma\big(r_t(x, y_1) - r_t(x, y_2)\big)\Big]^2$$
$$\leq \frac{1}{\sigma^2(1+R) \cdot \big(1 - \sigma(1+R)\big)^2} \cdot \frac{\log\big(|\Pi|/\delta\big)}{N},$$

where the first inequality holds due to $|a - b| \cdot |\max_{x \in [a,b]} \sigma'(x)| \leq |\sigma(a) - \sigma(b)|$ with the fact that $\sigma'(x) = \sigma(x)(1 - \sigma(x)) \leq \sigma(1+R)\big(1 - \sigma(1+R)\big)$ and the last inequality holds due to (E.2). Thus, we complete the proof of Lemma D.1. $\qquad\square$

### E.2 PROOF OF LEMMA D.2

*Proof of Lemma D.2.* We prove this inequality based on two situations: $x \geq 1$ or $x < 1$. For the case that $x \geq 1$, we have

$$(1+x) \cdot |\log x| = (1+x) \log x \geq (1+x)\left(1 - \frac{1}{x}\right) \geq x\left(1 - \frac{1}{x}\right) = x - 1,$$

where the first inequality holds due to $\log x \geq 1 - 1/x$ and the second inequality holds due to $x \geq 1$. We now consider the case where $x < 1$. In this case, we have

$$(1+x) \cdot |\log x| = -(1+x) \log x \geq -(1+x)(x-1) \geq -(x-1),$$

where the first inequality holds due to $\log x \leq x - 1$ and the second inequality holds due to $x < 1$. Combining the results in these two cases, we complete the proof of Lemma D.2. $\square$

### E.3 PROOF OF LEMMA D.3

In this section, we provide the proof of Lemma D.3, and we summarize our definitions and assumptions as follows. Similar assumptions are used in Rosset et al. (2024) to analyze the statistical error from the reparameterized reward.

**Definition E.2** (Feasible Policy Class). For each iteration $t \in [T]$, let $\Pi_t \subseteq \Pi$ represent the feasible policy class, which includes all potential policies $\pi_t$ generated by Algorithm 1 during iteration $t$, based on various possible samplings from the data collection process.

**Assumption E.3** (Realizability). For each iteration $t \in [T]$ and each potential policy $\pi \in \Pi_t$, we assume that the following updated policy also belongs to the policy class $\Pi$:

$$\widehat{\pi}(\cdot|x) = \frac{1}{Z_\pi(x)} \cdot \pi(\cdot|x) \cdot \exp\left(\frac{r_\pi^*(x,\cdot)}{\beta}\right) \in \Pi,$$

where $r_\pi^*(x,y) = \mathbb{E}_{y' \sim \pi}\left[\mathbb{P}(y \succ y'|x)\right]$ denotes the win probability against the behavior policy $\pi$ and $Z_\pi(x) = \sum_{y \in \mathcal{Y}} \pi(y|x) \exp\left(r_\pi^*(x,y)/\beta\right)$ is the partition function.

**Assumption E.4** (Boundedness). For each iteration $t \in [T]$ and each potential policy $\pi_t \in \Pi_t$, we have

$$\beta \log \frac{\widehat{\pi}_{t+1}(y|x)}{\pi_t(y|x)} \in [-R, R]$$

holds for all $x \in \mathcal{X}, u \in \mathcal{Y}$ and any potential collected data set $\mathcal{D}_t$.

Based on these assumption, we now begin the proof of Lemma D.3

*Proof of Lemma D.3.* For each policy $\pi \in \Pi_t$ and the corresponding reparameterized reward $r_\pi(x,y) = \beta \log\left(\pi(y|x)/\pi_t(y|x)\right)$, we have

$$\mathbb{E}_{x \sim \rho, y \sim \pi_t(\cdot|x)}\left[r_\pi(x,y) - r_t^*(x,y) + \log Z_{\pi_t}(x)\right]^2$$

$$= \mathbb{E}_{x \sim \rho, y \sim \pi_t(\cdot|x)}\left[r_\pi(x,y) - \mathbb{E}_{y' \sim \pi_t}\left[\mathbb{P}(y \succ y'|x)\right] + \log Z_{\pi_t}(x)\right]^2$$

$$= \mathbb{E}_{x \sim \rho, y, y' \sim \pi_t(\cdot|x)}\left[r_\pi(x,y) - \mathbb{1}(y \succ y'|x) + \log Z_{\pi_t}(x)\right]^2 + C_{\pi_t}$$

$$= \mathbb{E}_{x \sim \rho, y, y' \sim \pi_t(\cdot|x)}\left[r_\pi(x,y) - \mathbb{1}(y \succ y'|x) + \log Z_{\pi_t}(x)\right]^2/2$$

$$\qquad + \mathbb{E}_{x \sim \rho, y, y' \sim \pi_t(\cdot|x)}\left[r_\pi(x,y') - \mathbb{1}(y' \succ y|x) + \log Z_{\pi_t}(x)\right]^2/2 + C_{\pi_t}$$

$$= \mathbb{E}_{x \sim \rho, y^w, y^l \sim \pi_t}\left[\ell(r_\pi, x, y^w, y^l, \pi_t)\right] + C_{\pi_t},$$

where $Z_{\pi_t}(x) = \sum_{y \in \mathcal{Y}} \pi_t(y|x) \exp\left(r_t^*(x,y)/\beta\right)$ is the partition function, $C_{\pi_t} = \mathbb{E}_{x \sim \rho, y, y' \sim \pi_t(\cdot|x)}\left[\mathbb{P}(y \succ y'|x) - \mathbb{1}(y \succ y'|x)\right]^2$ is the variance of behavior policy $\pi_t$ and the third equation holds because $y, y'$ collected under the same behavior policy. Conditioned on Assumption E.3, there exists a policy $\pi_{t+1}^* \in \Pi$ such that $r_{\pi_{t+1}^*}(x,y) = \mathbb{E}_{y' \sim \pi_t}\left[\mathbb{P}(y \succ y'|x)\right] - \log Z_{\pi_t}(x)$. Under this situation, according to the standard concentration argument based on Bernstein's inequality, the least-squares estimator $\widehat{\pi}_{t+1}$ satisfies

$$\mathbb{E}_{x \sim \rho, y \sim \pi_t(\cdot|x)}\left[r_{\widehat{\pi}_{t+1}}(x,y) - r_{\pi_{t+1}^*}(x,y)\right]^2 \leq O\left(\frac{(R+1)^2 \cdot \log\left(|\Pi|/\delta\right)}{N}\right), \qquad (\text{E.3})$$

Table 4: **Open LLM Leaderboard Evaluation**. Comparison of APO with state-of-the-art iterative training algorithms, results are reported using `v0.4.1` of the `lm-evaluation-harness` library. Tasks and parameters follows Beeching et al. (2023).

| Models | Arc | TruthfulQA | WinoGrande | GSM8k | HellaSwag | MMLU | Avg. | Avg. 5 |
|---|---|---|---|---|---|---|---|---|
| Mistral-7B-Instruct-v0.2 | 63.74 | 66.81 | 77.90 | 42.00 | 84.77 | 59.12 | 65.72 | 70.47 |
| DPO Iter1 | 65.36 | 69.19 | 77.11 | 36.92 | 85.1 | 58.89 | 65.43 | 71.13 |
| DPO Iter2 | 66.47 | 69.93 | 78.14 | 42.46 | 85.44 | 58.29 | 66.79 | 71.65 |
| DPO Iter3 | 66.72 | 68.07 | 77.51 | 41.47 | 85.74 | 58.85 | 66.39 | 71.37 |
| APO Iter1 | 65.36 | 69.19 | 77.11 | 36.92 | 85.1 | 58.89 | 65.43 | 71.13 |
| APO Iter2 | 66.98 | 70.98 | 77.51 | 42.61 | 85.32 | 58.69 | **67.02** | **71.90** |
| APO Iter3 | 65.78 | 69.75 | 77.03 | 37.83 | 85.43 | 58.76 | 65.76 | 71.35 |

where the ineqaulity holds due to Lemma 6 in Xu & Zeevi (2020) with the Assumption E.4. Therefore, we have

$$
\mathbb{E}_{x\sim\rho,(y_1,y_2)\sim\pi_t}\left[\left(r_t^*(x,y_1) - r_t^*(x,y_2) - r_t(x,y_1) + r_t(x,y_2)\right)^2\right]
$$

$$
= \mathbb{E}_{x\sim\rho,(y_1,y_2)\sim\pi_t}\left[\left(r_{\pi_{t+1}^*}(x,y_1) - r_{\pi_{t+1}^*}(x,y_2) - r_t(x,y_1) + r_t(x,y_2)\right)^2\right]
$$

$$
\leq 2\mathbb{E}_{x\sim\rho,y_1\sim\pi_t}\left[r_{\pi_{t+1}^*}(x,y_1) - r_t(x,y_1)\right]^2 + 2\mathbb{E}_{x\sim\rho,y_2\sim\pi_t}\left[r_{\pi_{t+1}^*}(x,y_2) - r_t(x,y_2)\right]^2
$$

$$
\leq O\left(\frac{(R+1)^2 \cdot \log\left(|\Pi|/\delta\right)}{N}\right),
$$

where the first inequality holds due to $(x+y)^2 \leq 2x^2 + 2y^2$ and the last inequality holds due to (E.3). Thus, we complete the proof of Lemma D.3. $\qquad\square$

# F  ADDITIONAL DETAILS IN EXPERIMENTS

## F.1  HYPERPARAMETERS

Our training is conducted on eight A100 GPUs, setting a global batch size of 64. This is done through setting a local batch size of 8 across all GPUs, with a gradient accumulation step of 1. We utilize the RMSProp optimizer for each iteration, tuning the learning rate and $\beta$ parameters. Although we initially considered a range of learning rates from $[5 \times 10^{-7}, 1 \times 10^{-7}, 5 \times 10^{-8}]$, we found that a fixed learning rate of $5 \times 10^{-7}$ while adjusting $\beta$ across iterations results in enhanced performance. The final $\beta$ value is set at 0.01 for the first iteration, 0.1 for the second iteration, and 1.0 for the third iteration. The extrapolation parameter $\alpha$ is set to be a constant 0.3. In addition, the learning rate follows a linear schedule with a warm-up ratio of 0.1. For inference, we employ the `vllm` library (Kwon et al., 2023) for response generation, configured with a temperature of 0.7 and a top_p of 0.9. The maximum token length for response generation is set to 2048 tokens. Finally, evaluations across all benchmarks are conducted using eight A6000 GPUs.

## F.2  OPEN LLM LEADERBOARD EVALUATION

We provide the result of Open LLM Leaderboard. We calculate the average scores across six tasks, as well as the average of five tasks excluding GSM8k. This exclusion is due to our training's emphasis on general instruction-following ability, with no exposure to correct answers for math problems. Table 4 shows that APO demonstrates robust performance across multiple tasks. Notably, at iteration 2, APO achieves the highest overall average score of 67.02, outperforming the baseline models. When focusing on the average of five key tasks, excluding GSM8k, APO iteration 2 again leads with an impressive score of 71.90. Although APO iteration 3 shows a slight decrease in the overall average score, it maintains a competitive edge. This decline is primarily attributed to the low correlation of multiple-choice question tasks with instruction-following abilities. we note that all trained baselines exhibit similar performance suboptimality on this metric, while the AlpacaEval performance increases.

## F.3  RESPONSE EXAMPLES IN DIFFERENT ITERATIONS

Table 5: Generation example of our fine-tuned model by APO at different iterations.

| Prompt | Hi, my sister and her girlfriends want me to play kickball with them. Can you explain how the game is played, so they don't take advantage of me? |
|---|---|
| Iteration 1 | Hello! I'd be happy to help explain the rules of kickball to ensure that you feel confident and prepared for your game with your sister and her girlfriends. Kickball is a fun, social, and inclusive game that combines elements of baseball and soccer. Here's a breakdown of the essential rules: 1. Teams and Setup: - Divide into two teams, ideally of equal numbers. - Designate a field with a home base, a first base, a second base, a third base, and a kickball diamond. - Choose a team to be in the field (defense) and one to be at bat (offense). 2. Equipment: - A standard soccer ball (or a large, soft kickball) is used. - No specialized equipment is required for players, except for comfortable clothing and athletic shoes. 3. Objective: - The objective of the game is for the team at bat to score runs by kicking the ball and advancing around the bases, while the fielding team tries to prevent this by getting the runners out. 4. Batting: - The game begins with a pitch (roll) from the pitcher (field player) to the kicker (batter). - The kicker attempts to kick the ball with their foot (no toe-kicks or using hands are allowed). - After a valid kick, the runner(s) may advance around the bases, trying to reach the next base before the ball reaches that base or a defensive player tags them. 5. Fielding: - The fielding team's goal is to get the runners out. - Players can attempt to catch a kicked ball in mid-air to retire the side (three consecutive outs), or they can touch a base with the ball before a runner reaches it to force them out. 6. Scoring: - A run is scored when a player advances around all four bases and returns to home plate. - Each inning is typically limited to a specific number of outs or runs scored (agree on this beforehand). 7. Fair Play: - Encourage sportsmanship and ensure all players understand and respect the rules to maintain a fun and enjoyable experience for everyone involved. 8. Additional rules: - Discuss any specific house rules or modifications to the game before starting, such as allowing multiple kicks per at-bat or implementing a mercy rule if one team is significantly ahead in runs. I hope this information is helpful! If you have any further questions or need clarification on any rules, feel free to ask. Good luck and have a great time playing kickball with your sister and her girlfriends! |

Table 6: Generation example of our fine-tuned model by APO at different iterations.

| Prompt | Hi, my sister and her girlfriends want me to play kickball with them. Can you explain how the game is played, so they don't take advantage of me? |
|---|---|
| Iteration 2 | Hello! It's great that your sister and her friends have invited you to join them for a game of kickball. Kickball is a fun, social, and inclusive team sport that combines elements of baseball and soccer. Here's a basic explanation of how the game is played to ensure you feel prepared and confident on the field. Keep in mind that variations may exist depending on the specific rules your group chooses to follow, but this outline should provide a solid foundation. 1. Teams and Setup: Divide into two even teams, ideally with similar numbers of players (typically 8-12 per team). Designate one team as the home team (they'll be up to bat first), and the other as the visiting team. Set up a diamond-shaped kicking area (the "kickline") approximately 60 feet from a base diamond, similar to baseball. Mark out four bases (first, second, third, and home) in a square shape around the diamond, with home plate located at the end of the kickline. 2. Objective: The objective of kickball is to score runs by hitting the ball with your foot and advancing around the bases before returning to home plate safely. The opposing team aims to get outs by fielding the ball and tagging runners with the ball or touching them with it while they're not on a base. 3. Batting: The batter stands at home plate and kicks the ball when it's pitched towards them by the pitcher from the opposing team. A runner may advance from their base at the start of each pitch, but they must return to that base before the ball is touched by a fielder or thrown to that base. If a runner advances too far before returning to their base, they risk being tagged out in a force play. 4. Fielding: Fielders position themselves around the diamond and outfield to catch fly balls or field grounders. They can also attempt to tag runners with the ball while they're advancing between bases or when they reach a base. Once a fielder catches a fly ball or tags a runner, they record an out. A maximum of three outs per inning is typical. 5. Scoring: Runners score by advancing around the bases and returning safely to home plate. A run is scored when a player reaches home plate after kicking the ball and successfully advances around all four bases without being tagged out or reaching base before a preceding runner is retired (resulting in that runner being forced out). 6. Inning Structure: Each team gets an opportunity to bat in each half-inning. The inning ends when three outs are recorded by the fielding team. Typically, games consist of multiple innings until one team reaches a predetermined number of runs (e.g., five or ten). 7. Fair Play: Ensure that everyone follows the rules and plays in good spirits. Kickball is meant to be an enjoyable experience for all involved, so focus on having fun while staying mindful of safety and fairness. If you feel unsure about any aspect of the game or suspect your sister and her friends might be taking advantage of you, communicate openly with them about your concerns and work together to find solutions that maintain a balanced and enjoyable playing field for everyone. |

Table 7: Generation example of our fine-tuned model by APO at different iterations.

| Prompt | Hi, my sister and her girlfriends want me to play kickball with them. Can you explain how the game is played, so they don't take advantage of me? |
|---|---|
| Iteration 3 | Hello! It's great that your sister and her friends have invited you to join their kickball game. Kickball is a fun, social, and inclusive sport, well-loved for its simplicity and energy. To ensure you feel confident and prepared for the game, here's a basic explanation of how kickball rules typically work. Keep in mind that variations exist, so always clarify any specific rules with your group before playing. 1. Teams and Setup: Divide into two teams, ideally through an evenly distributed random draw or team selection process. Designate a field with clear boundaries using cones or markers Ž2013 a diamond shape or rectangle works well. Each team defends a designated home base (typically marked by a large cone or object). 2. Equipment: A standard soccer ball is commonly used for kickball games. No specialized equipment is required beyond that, although gloves may be worn for comfort or to improve grip on the ball. 3. Objective: The objective of the game is to score points by kicking the ball and successfully advancing around the bases Ž2013 first base, second base, third base, and home plate Ž2013 before returning to home base. A run is scored when a player reaches home base after completing a full circuit around the bases. 4. Starting the Game: A coin toss determines which team bats first. The fielding team positions themselves around the bases and in the outfield, ready to receive the kicked ball. The batting team gathers near home plate with a designated kicker ready to start the inning. 5. Kicking and Running: The kicker takes a stance and kicks the ball forcefully with their foot, aiming for a solid contact to launch the ball as far as possible. Runners on base attempt to advance based on where the ball lands and how the fielding team reacts. 6. Fielding Rules: Fielders must retrieve or touch the ball before a runner advances to the next base. Once a fielder has the ball, they can tag runners with it to record outs. However, runners can advance on a missed catch or an overthrow. If a fielder touches a base before a runner arrives, the runner is out. 7. Scoring: A run is scored when a batter safely kicks the ball and reaches first base, while teammates advance around the bases and return to home plate before the inning ends (three outs). 8. Inning Structure: Each team takes turns at-bat and in the field. Typically, an inning lasts until the fielding team records three outs, at which point the teams switch roles. A game consists of multiple innings, with the team with the most runs at the end declared the winner. 9. Fair Play and Sportsmanship: Encourage a friendly and respectful environment, following the spirit of the game. Ensure all players understand the rules, and focus on having fun rather than taking advantage of one another. If needed, consult your group to establish additional guidelines or modifications to accommodate skill levels and preferences. 10. Additional Tips: - Wear comfortable clothing and shoes appropriate for running and kicking. - Stay hydrated and take breaks as needed. - Communicate clearly with teammates and opponents. - Focus on improving your skills, both as a kicker and a runner, to contribute effectively to your team. With this foundational knowledge, you should feel more confident joining your sister and her friends for a kickball game. Enjoy the experience and have fun! |

Table 8: Another generation example of our fine-tuned model by APO at different iterations.

| Prompt | How did US states get their names? |
|---|---|
| Iteration 1 | The naming of US states reflects a rich and varied history, influenced by explorers, Native American cultures, natural features, and political considerations. Here are some common ways US states got their names: 1. Names derived from Native American words: Many states' names come from Native American languages or place names. For example, California comes from the Spanish rendition of the Native American word "Quercus californius," meaning "californio," or "californian," which referred to the native people or the abundant california trees. Florida's name comes from the Spanish word "Pascuan Floridus," meaning "flowery land," a term used by explorer Juan Ponce de LeŎ0f3n, likely inspired by the abundant natural beauty he encountered. 2. Names inspired by explorers and European settlers: Some states are named after famous explorers, settlers, or colonial founders. For instance, Virginia was named after Queen Elizabeth I of England, who was known as the "Virgin Queen" due to her status as an unmarried monarch. Georgia was named after King George II of Great Britain in honor of his support for the founding of a colony there. 3. Names based on geographical features: Several states are named after prominent geographical features, such as mountains, rivers, or bodies of water. For example, Colorado's name comes from the Spanish word "colorado," meaning "red," a term often used to describe the reddish hue of the area's soil and water. Montana's name means "mountainous" in Latin. 4. Names reflecting historical events or early settlement patterns: Some states' names are connected to significant historical events or early settlement patterns. For instance, Maine was originally called "Norumbega" based on tales of a mythical northern land of abundance, but was later renamed by English explorer John Smith to reflect its northern location relative to Massachusetts. Texas was named by the Spanish as "Tejas," meaning "friends" or "allies," due to the friendly reception given to Spanish explorers by the local Caddo and Tonkawa tribes. 5. Names honoring political or ideological affiliations: A few states' names reflect political or ideological affiliations. For example, Vermont was originally called "New Vermont" or "New England's Plymouth" due to its ties to the English county of Vermont and the Pilgrims of Plymouth Colony. Later, when it sought independence from New York, the name was adopted as a tribute to the revolutionary ideals of the Republic of Vermont. Overall, US states' names offer a fascinating glimpse into the rich history of exploration, settlement, and political development that shaped the American nation. |

Table 9: Another generation example of our fine-tuned model by APO at different iterations.

| Prompt | How did US states get their names? |
|---|---|
| Iteration 2 | US states have varied histories regarding how they received their names, with some derived from native place names or geographical features, others from explorers or colonial founding figures, and still others from natural elements or symbolic concepts. Here's a brief overview of how some common state names came to be: 1. Alabama: Derived from the Native American Chickasaw and Creek words "alhahoma" or "allapaha," meaning "thicketed place" or "place with ticklenut trees." 2. Alaska: Russian explorer Vitus Bering, who first documented the region in 1741, named it "Alaska" from the Russian words "alas kŏ430ŏ44f," meaning "great land" or "large country." 3. Arizona: Named after the Arizona Mining District, which in turn was derived from the Spanish term "Arizonia," meaning "arid" or "dry," likely in reference to the desert climate. 4. Arkansas: Originally called the "Land of Hieroglyphics" by early European explorers due to the abundance of Native American petroglyphs found in the region. Later, it was named Arkansas after the Quapaw tribe's name for a village near present-day Osceola, which translates to "people of the south wind." 5. California: Spanish explorer Juan Rodriguez Cabrillo is credited with giving the name "California" to the region in 1542, possibly inspired by the ancient mythical island of Califa described by Greek geographer Ptolemy as lying offshore west of the Pacific Ocean, abundant with gold and paradisiacal conditions. 6. Colorado: Derived from the Spanish word "colorado," meaning "red," likely in reference to the red soil and red sandstone formations found in the region. 7. Connecticut: Named by Dutch explorer Adriaen Block in honor of the Mohegan and Montauk Sachems (tribal leaders) Huskera or Quinipet and Sinigtot or Sitenico, whom he met in 1614. The name Connecticut comes from the Mohegan word "quinnipiac," meaning "long tidal river." 8. Delaware: Named after Sir Thomas West, 3rd Baron De La Warr, an English nobleman who served as the first colonial governor in 1630. The name Delaware comes from the Delaware River, which was named after the Lenape Native American tribe, whose name means "upstream people" or "people of the cleared land." 9. Florida: Spanish explorer Juan Ponce de Leŏ0f3n is credited with discovering Florida in 1513 and naming it after "Pascua Florida," meaning "feast of flowers," likely due to the abundant blooming vegetation he encountered during his expedition. 10. Georgia: Named after King George II of Great Britain in honor of his support for establishing a colony there in 1732. 11. Hawaii: The islands' indigenous Polynesian name, "Hawaii Loa," translates to "long home" or "homeland," reflecting the archipelago's unique identity as a distinct and unified cultural entity. The name Hawaii became widely known to Europeans through British explorer James Cook, who visited in 1778 and adopted it for the larger island of Hawaii in his charts. 12. Idaho: Derived from a Shoshone Native American word, "hee-da-ho," meaning "gem of the mountains" or "the sun comes from the waters." This name was popularized during the late 19th century when Idaho was being considered for statehood. 13. Illinois: Derived from a Native American word, "Illiniwek," meaning "tribe of superior men" or "tribe dwelling properly." This term referred to various tribes within the Illini Confederacy, which inhabited much of present-day Illinois prior to European contact. 14. Indiana: Named after the Native American term "Indianŏ0e1," meaning "land of the Indians," reflecting that the region was inhabited by various Native American tribes when European settlers arrived. 15. Iowa: Derived from a Native American word, "ayoua," meaning "prairie" or "beautiful land." This name was popularized during the late 19th century when Iowa was being considered for statehood. These examples represent just a fraction of US states and their unique origin stories. Many other states have names rooted in Native American languages, European explorers, geographical features, or symbolic concepts. |

Table 10: Another generation example of our fine-tuned model by APO at different iterations.

| Prompt | How did US states get their names? |
|---|---|
| Iteration 3 | US states have varied histories regarding how they earned their names. Here are some common ways and specific examples: 1. Indigenous Place Names: Many state names originate from Native American languages spoken by tribes inhabiting the regions prior to European exploration. For instance: - Florida: Derived from the Native American word "Pvscaleola " or "Scal scal lla, " meaning "flowing water " or "land of flowers, " depending on the specific indigenous tribe. European explorer Juan Ponce de Leon may have chosen this name based on reports of native peoples and their designations for the land. - Missouri: Derived from the Missouri River, which in turn took its name from the Miwoksi or Missouri Nation, a Siouan-speaking people inhabiting the area at the time of French explorer Louis Jolliet and Jacques Marquette's passage in 1673. Their name translates roughly to "people of large canoes. " 2. European Explorers and Settlers: Many states were named after early European explorers, settlers, or monarchs. For example: - Virginia: Named after Queen Elizabeth I of England in honor of her reign and patronage, specifically her title "the Virgin Queen. " Sir Walter Raleigh, who received a royal charter to establish colonies in North America, chose this name. - California: Named by Spanish explorer Vasco NŎ0faŎ0f1ez de Balboa in 1542, who gave it the Latinized name "California " after the Classical mythological figure Californias, who was believed to inhabit a paradisiacal island on the western edge of the known world. 3. Natural Features: Some states bear names inspired by distinctive geographic or topographical features. For instance: - Colorado: Derived from the Spanish "colorado, " meaning "red " Ž013 a reference to the red soil and rock formations found throughout the region. - Mo ntana: Named for the Latin word "montanus, " meaning "mountainous. " 4. Historical Events: A few states have names connected to significant historical events. For example: - Texas: Derived from the Spanish "Tejas, " meaning "friends " or "allies. " The name referred to the indigenous peoples who initially welcomed Spanish explorers in the late 16th century. Later, Mexico used the name when it ruled the region, and after Texas gained independence in 1836, the name remained. - Wisconsin: Named by French explorer Jean Nicolet in 1634 after the native peoples he encountered, the "Wisconsin " or "Meskousing " Ž013 an Algonquian-speaking people. The name "Wisconsin " is believed to mean "gathering place " or "place of the red people. " 5. Symbolic Meanings: In some cases, state names reflect symbolic or metaphorical concepts. For example: - Nevada: Derived from the Spanish "nevado, " meaning "snow-covered mountain range, " which aptly describes the state's mountainous terrain and its name's association with the silver rush that brought settlers to the area in the late 19th century (the silver being referred to as "white gold " or "silver snow "). These examples illustrate just a few ways US states have acquired their names throughout history. The specific origins of each state name are complex and often intertwined with the rich tapestry of indigenous, European, and American cultural influences that have shaped the United States. |