# OpenReview forum: "Accelerated Preference Optimization for Large Language Model Alignment"
_ICLR.cc/2025/Conference — Submitted to ICLR 2025_

### Official Review · Reviewer_A4Tx · 2024-10-23

**Soundness:** 2
**Presentation:** 3
**Contribution:** 2
**Rating:** 5
**Confidence:** 3

**Summary:**

This paper focuses on reinforcement learning from human feedback (RLHF) for large language models (LLMs), specifically examining the iterative preference optimization framework. It first observes that existing methods within this framework resemble the proximal point method in the optimization literature. Building on this connection, the paper proposes the accelerated preference optimization (APO) framework. APO incorporates Nesterov's momentum technique, originally introduced in optimization, to accelerate the preference optimization process. Theoretically, it proves that APO achieves a faster convergence rate compared to existing methods. Experimentally, the authors demonstrate APO's superiority over DPO, iterative DPO, and other preference optimization methods on the AlpacaEval 2.0 benchmark.

**Strengths:**

1. The paper establishes a novel and intriguing connection between iterative preference optimization and proximal point optimization. This connection has the potential to inspire further research that leverages advanced techniques from classical optimization literature for preference optimization.
2. The paper presents a rigorous theoretical analysis of the proposed APO method, demonstrating that APO can achieve a faster convergence rate.

**Weaknesses:**

1. The theoretical results in this paper are not particularly strong and do not conclusively show that APO is superior to previous methods. Specifically, in Theorem 4.4, compared to DPO ($\alpha=0$), APO achieves a smaller optimization error by a factor of $1-\alpha$, but incurs a larger statistical error by a factor of $1/(1-\alpha)$. Therefore, in the finite-data case, it remains unclear whether APO is theoretically better than DPO. Similar concerns also arise in Theorem 4.8.
2. There is a mismatch between the theoretical results and the experiments in this paper. Theoretically, the paper proves that APO achieves a convergence rate of the order $\widetilde{\mathcal{O}}((1-\alpha) / t)$. This rate is meaningful only in the large-iteration regime, as it ignores certain constants. However, in the experiments, the LLM is trained with only a small $t=3$. This discrepancy creates a gap between the theoretical analysis and the empirical results. As a result, I believe that the improved convergence rate in theory does not fully explain APO's empirical performance, and the experimental results do not adequately reflect its theoretical convergence rate.

**Questions:**

In Equation (3.1), the loss function for reward models is denoted as $\ell\left(r, x, y^w, y^l, \pi_t\right)$. However, it appears that this function does not actually take $\pi_t$ as an input.

---

> ### Author Response · Authors · 2024-11-24
> **Response to Reviewer A4Tx**
>
> Thanks for your valuable feedback!
>
> ----
>
> **Q1**: The theoretical results in this paper are not particularly strong and do not conclusively show that APO is superior to previous methods. Specifically, in Theorem 4.4, compared to DPO (α=0), APO achieves a smaller optimization error by a factor of 1−α, but incurs a larger statistical error by a factor of 1/(1−α). Therefore, in the finite-data case, it remains unclear whether APO is theoretically better than DPO. Similar concerns also arise in Theorem 4.8.
>
> **A1**: Thank you for highlighting this trade-off. We agree that adding momentum increases statistical error. However, as noted in Remark 4.5, when working with sufficiently large datasets, which is widely studied in the analysis of preference-based optimization method, the optimization error becomes the dominant term in determining the sub-optimality gap. In these practical cases, our algorithm's ability to reduce this dominant term by a factor of $(1−α)$ represents a meaningful improvement. This improvement is particularly significant because in many real applications (e.g., $N=2K$ samples and $T=O(1)$ iterations in our experiments), the reduction in optimization error outweighs the increase in statistical error. Our theoretical analysis thus provides valuable insights into how momentum can be effectively leveraged to enhance convergence in preference-based optimization, especially when working with substantial amounts of preference data. This is also backed up by our empirical experiment results.
>
> ----
>
> **Q2**: There is a mismatch between the theoretical results and the experiments in this paper. Theoretically, the paper proves that APO achieves a convergence rate of the order O~((1−α)/t). This rate is meaningful only in the large-iteration regime, as it ignores certain constants. However, in the experiments, the LLM is trained with only a small t=3. This discrepancy creates a gap between the theoretical analysis and the empirical results. As a result, I believe that the improved convergence rate in theory does not fully explain APO's empirical performance, and the experimental results do not adequately reflect its theoretical convergence rate.
>
> **A2**: We appreciate this insightful observation regarding the gap between our theoretical analysis and the current empirical setup. Increasing the number of iterations $T$ improves performance but also leads to higher computational time and cost, requiring a careful balance between the benefits and the additional overhead. To address this, we are conducting further experiments with more iterations to better validate the theoretical convergence rate and will update our response with the results as soon as they are available.
>
>
> ----
>
> **Q3**: In Equation (3.1), the loss function for reward models is denoted as $ℓ(r,x,y^w,y^l,π_t)$. However, it appears that this function does not actually take πt as an input.
>
> **A3**: In our work, we consider the general form of loss functions. While DPO loss does not depend on $π_t$, other loss functions like SPPO (Example 3.2) do utilize $π_t$ in their formulation. Therefore, we maintain $π_t$ in the general loss function notation to accommodate these cases. This generalized notation allows our framework to encompass a broader class of preference-based optimization methods, including those where the loss function explicitly depends on the current policy. By maintaining this general form, our theoretical analysis and algorithmic framework can be applied to various preference-based optimization scenarios, providing flexibility in the choice of loss functions while maintaining theoretical guarantees.
>
>
> ----

---

> > ### Comment · Reviewer_A4Tx · 2024-11-27
> > **Thank you for the detailed response!**
> >
> > **Regarding the trade-off between statistical error and optimization error:**
> > I respectfully disagree with the claim that "when working with sufficiently large datasets, which is widely studied in the analysis of preference-based optimization methods, the optimization error becomes the dominant term in determining the sub-optimality gap." In many theoretical works on preference-based methods [1, 2, 3], the statistical error is treated as the primary error term in the sub-optimality gap and is often associated with reward hacking issues. Therefore, the increase in statistical error caused by introducing momentum should be carefully considered.
> >
> > **Regarding the mismatch between theory and practice:**
> > My concern regarding the mismatch between theory and practice still stands, and I look forward to further experimental results.
> >
> > ---
> >
> > ### References:
> > [1] Principled Reinforcement Learning with Human Feedback from Pairwise or K-wise Comparisons
> > [2] Exploratory Preference Optimization: Harnessing Implicit Q*-Approximation for Sample-Efficient RLHF
> > [3] Correcting the Mythos of KL-Regularization: Direct Alignment without Overoptimization via χ²-Preference Optimization

---

> > > ### Author Response · Authors · 2024-11-28
> > > **Response to Reviewer A4Tx's feedback**
> > >
> > > Thanks for your further feedback. We have provided answers to each of your points:
> > >
> > > ----
> > > **Q1**: Regarding the trade-off between statistical error and optimization error:
> > > I respectfully disagree with the claim that "when working with sufficiently large datasets, which is widely studied in the analysis of preference-based optimization methods, the optimization error becomes the dominant term in determining the sub-optimality gap." In many theoretical works on preference-based methods [1, 2, 3], the statistical error is treated as the primary error term in the sub-optimality gap and is often associated with reward hacking issues. Therefore, the increase in statistical error caused by introducing momentum should be carefully considered.
> > >
> > > **A1**: Thanks for pointing out these related works. After careful examination, we want to clarify that optimization error only exists in iterative optimization processes and does not appear in single-iteration methods. When the optimization process involves multiple iterations to approach the optimal policy, the optimization error becomes the dominant term if \textbf{the number of iterations is much smaller than the dataset size}. Here are our detailed comments on these related works:
> > >
> > > [1]: This work studied the Pessimistic MLE algorithm for RLHF with single iteration, which provides a confidence region for the underlying model and avoids iterative preference learning. Without an iteration optimization process, statistical error is the only considered term here. However, their approach relies heavily on special structure (linear in this work) and is difficult to implement in LLM applications.
> > >
> > > [2]: In this work, the algorithm collects only a single preference pair per iteration. Even when reusing previously collected data, the dataset size in preference optimization is $O(T)$. In this regime, the statistical error (scaling with $\sqrt{1/N}$) dominates the optimization error (scaling with $1/T$ or $\exp(-T)$).
> > > However, in real-world applications like SPPO and Iterative KTO, we typically have few iterations but large datasets (e.g., $N=20K$ samples). In this practical regime, where $N>>T$, the optimization error becomes dominant due to large dataset.
> > >
> > >
> > > [3]: This work proposes two different algorithms:
> > >
> > > 1:  Algorithm 1 focuses on finding an optimization policy with a KL-regularization term for the deviation from a reference policy, which only has a single iteration. In comparison, our algorithm aims to identify the actual optimal policy through Bregman Proximal Point Method with iterative preference optimization, where each iteration corresponds to an optimization policy with different reference policy. Due to this different goal and algorithm structure, the optimization error only exists during the iteration learning process and there is no such error in Algorithm 1 with a single iteration process, and hence it only focuses on statistical error.
> > >
> > > 2: Algorithm 2 faces a similar case to [2], where the algorithm only collects a single preference pair in each iteration, leading to dataset size in preference optimization up to $N=O(T)$. Thus, the statistical error becomes the dominant term.
> > >
> > >
> > > ----
> > >
> > > **Q2**: My concern regarding the mismatch between theory and practice still stands, and I look forward to further experimental results.
> > >
> > > **A2**: During the rebuttal period, we have added experiments for:
> > >
> > > 1. Additional baselines in the empirical studies - IPO, SPPO, SimPO, KTO and ExPO
> > >
> > > 2. Comparative experiments between AdamW and RMSprop optimizers
> > >
> > > Due to time limitations, we are still working on conducting further experiments with more iterations to better validate the theoretical convergence rate and will update our response with these results as soon as they become available.

---

### Official Review · Reviewer_EXfT · 2024-11-01

**Soundness:** 3
**Presentation:** 4
**Contribution:** 3
**Rating:** 5
**Confidence:** 3

**Summary:**

DPO is a popular offline methodology for LLM alignment. It has led to a family of preference optimization algorithms with different formulations of loss functions and problem formulations. In this work, the authors formulate iterative preference optimization (a variant of DPO class of algorithms) as an optimization problem with proximal point method being the optimization framework. Consequently, the authors use Nesterov's momentum technique to speed up the convergence.

The authors demonstrate theoretically that the new framework (APO) leads to faster converge in comparison to standard iterative preference optimization methods. They also conduct empirical experiments to demonstrate the improvements of APO

**Strengths:**

1. To the best of my knowledge, this is the first work that examines the converge of iterative DPO and proposes improvements towards the convergence specifically
2. The theoretical analysis demonstrates that APO does lead to faster convergence in comparison to standard DPO
3. The empirical results shows that APO leads to better alignment in comparison to the baselines.

**Weaknesses:**

1. To the best of my understanding, Equation 3.6 in the APO algorithm does not update the neural network weights. Instead it is derived by keeping reference of the policy from the previous and the current iterations. It is not clear how the final policy in step 7 is used in the next iterations in steps 5 and 6.
2. This work might benefit from additional baselines in the empirical studies - IPO, SPPO, SimPo, KTO and especially ExPO
3. To the best of my understanding, the extrapolation step adds additional memory overhead since it keeps a reference of two iterations of the policy. This might not be practical for very large language models.

**Questions:**

N/A

---

> ### Author Response · Authors · 2024-11-24
> **Response to Reviewer EXfT**
>
> Thanks for your helpful suggestions and comments!
>
> ----
>
> **Q1** To the best of my understanding, Equation 3.6 in the APO algorithm does not update the neural network weights. Instead it is derived by keeping reference of the policy from the previous and the current iterations. It is not clear how the final policy in step 7 is used in the next iterations in steps 5 and 6.
>
> **A1**  First, we want to clarify that in implementation, we perform updates on parameters rather than keeping reference of the policy from the previous and current iterations. As we discussed in Section 5.2, when the policy is a softmax linear function, this update reduces to a momentum extrapolation in the parameter space. Therefore, we implement this momentum step by performing the extrapolation directly in the neural network parameter space to approximate the corresponding momentum update (3.6) that is defined in the probability space. Second, for the next iteration, the final policy in step 7 plays two roles: it serves as the reference policy for collecting new preference data and also sets the reparameterized function class for optimization.
>
> ----
>
> **Q2** This work might benefit from additional baselines in the empirical studies - IPO, SPPO, SimPo, KTO and especially ExPO
>
> **A2** We have conducted comprehensive experiments with several baselines. Here are the detailed results:
>
> | Model | AlpacaEval 2.0 LC Win Rate | AlpacaEval 2.0 Win Rate | Std Dev | Length |
> |------------|----------------------------|-------------------------|---------|--------|
> | KTO iter 1 | 23.81 | 20.88 | 1.23 | 1710 |
> | KTO iter 2 | 21.02 | 19.85 | 1.17 | 1859 |
> | KTO iter 3 | 23.76 | 23.84 | 1.27 | 1972 |
>
> | Model | AlpacaEval 2.0 LC Win Rate | AlpacaEval 2.0 Win Rate | Std Dev | Length |
> |------------|----------------------------|-------------------------|---------|--------|
> | SPPO iter 1 | 26.36 | 24.04 | 1.28 | 1802 |
> | SPPO iter 2 | 28.38 | 27.43 | 1.32 | 1909 |
> | SPPO iter 3 | 29.17 | 29.75 | 1.37 | 2051 |
>
> | Model | AlpacaEval 2.0 LC Win Rate | AlpacaEval 2.0 Win Rate | Std Dev | Length |
> |------------|----------------------------|-------------------------|---------|--------|
> | IPO iter 1 | 25.04 | 23.36 | 1.27 | 1843 |
> | IPO iter 2 | 26.78 | 30.66 | 1.38 | 2315 |
> | IPO iter 3 | 28.07 | 37.03 | 1.43 | 3153 |
>
> | Model | AlpacaEval 2.0 LC Win Rate | AlpacaEval 2.0 Win Rate | Std Dev | Length |
> |------------|----------------------------|-------------------------|---------|--------|
> | ExPO on DPO iter 3 | 18.83 | 31.21 | 1.38 | 3810 |
>
> Compared to our results in Table 1, these results highlight the superior performance of our method over these baselines.

---

### Official Review · Reviewer_Nq8c · 2024-11-02

**Soundness:** 2
**Presentation:** 3
**Contribution:** 2
**Rating:** 5
**Confidence:** 4

**Summary:**

This paper studies the optimization problem in the reinforcement learning from human preferences setup. It focuses on the direct preference optimization algorithm and proposes using the classical idea of Nesterov's acceleration. The paper theoretically proves that the algorithm reduces optimization sub-optimality but at the cost of increasing the statistical generalization error. Numerical results on the Mistral model with the UltraFeedback dataset are provided.

**Strengths:**

1. This paper is well-written and easy to follow.
2. It provides a new theory for validating Nesterov's extrapolation idea in the iterative DPO framework.

**Weaknesses:**

1. There is a trade-off between optimization error and statistical error in the developed theory. Therefore, it is unclear whether Nesterov's acceleration offers a true advantage over the naive method.
2. Empirically and theoretically, the advantages and weaknesses compared to direct reward optimization algorithms (e.g., PPO) are unclear.
3. Experiment results are somewhat weak and does not validate the theory.

**Questions:**

1. The implementation of Algorithm 1 is unclear.

    - To the reviewer’s understanding, new preference data is required in each iteration $t = 0, 1, \dots, T$. Is this correct? Could the authors clarify the preference collection process? Is the preference collected using the BT model with random noise?

    - Related to this, in the experiments, a reward model is used. Why not study direct reward maximization (e.g., PPO) both empirically and theoretically? To the reviewer's knowledge, direct reward maximization methods can handle such tasks effectively and demonstrate strong empirical evidence in practice.

    - How should (3.6) be implemented in practice? Could the authors provide more details? Should the parameter or the distribution be extrapolated? Additionally, how can this result be extended to cases where $ y $ is sequential? Is it straightforward to apply this closed-form solution?

2. The reviewer is concerned about the soundness of the theory.

    -  Assumption 4.3 is strong. Essentially, it requires $ \pi_t $ to be a purely stochastic policy. On the other hand, the optimal policy $ \pi^* $ in Theorem 4.4 is deterministic. Therefore, the theory with this assumption seems problematic in the asymptotic case.

    -  The implications of the theory are based on upper bounds and are weak. Could you provide simulations for a simple case to validate the tightness of the developed method? Additionally, could you use simulations to study the trade-off between optimization sub-optimality and statistical error?

      -  It is recommended to explain in the main text why momentum increases the statistical error.

3. Comments on experimental settings and results:

    -  In the experiments, it seems unusual to have 2k instructions per iteration. To the reviewer's knowledge, the entire dataset contains 64k samples. Could the authors explain the rationale behind this choice? Furthermore, why are the experiments conducted with only three iterations? Could performance continue to improve with more iterations?

    -  Why was the RMSprop optimizer used in the experiments instead of the Adam optimizer? The Adam optimizer includes momentum, which can also accelerate optimization, similar to Nesterov's acceleration technique. The reviewer would like to see empirical results using the Adam optimizer to determine whether the proposed technique shows a clear advantage in this scenario.

    -  The improvements on AlpacaEval and MT-Bench are promising but come with some concerns. Evaluation results on these two benchmarks can be manipulated (e.g., by increasing response length). The reviewer appreciates that the length-controlled win rate is used; this is a positive step. However, concerns remain, as the training data (ultrafeedback) is collected by GPT-4 and evaluation on these benchmarks also relies on GPT-4. Could you test your method on other, more reliable benchmarks, such as IFeval or math reasoning tasks?

    -  Empirically, the advantage of this approach over the model extrapolation technique introduced by Zheng et al. (2024a) is unclear. Additionally, the empirical comparison with direct reward optimization methods is also unclear, which is important for establishing the superiority of the proposed method and techniques.

---

> ### Author Response · Authors · 2024-11-24
> **Response to Reviewer Nq8c (Part 1)**
>
> Thank you for your careful review and detailed feedback!
>
> ----
>
> **Q1**: Assumption 4.3 is strong. Essentially, it requires $π_t$ to be a purely stochastic policy. On the other hand, the optimal policy $π^*$ in Theorem 4.4 is deterministic. Therefore, the theory with this assumption seems problematic in the asymptotic case.
>
> **A1**: It is very common to use stochastic policy in RL algorithms even though there always exists a deterministic optimal policy. For example, in one of the most widely used RLHF algorithms, PPO, the output policy is also stochastic. From the theoretical perspective, we want to clarify that our realizability does not require  $π^*$  to fall into the function class, and the stochastic policy cannot fully match the deterministic optimal policy  $π^*$ . Nevertheless, according to our Theorem 4.8, we have shown that the TV-distance between the stochastic policy $\hat{\pi}_{t+1}$ and the optimal policy $\pi^*(x)=\arg \max _{y\in Y} r^*(x,y)$ will converge to 0 after enough iterations ($T$) and samples ($N$), and this holds exactly in the asymptotic sense. In addition, as we stated before Assumption 4.3, similar assumptions have also been used in the previous analysis of preference-based optimization method [1]. We believe this is a standard assumption and not strong at all. We hope the explanation above helps to clarify any misunderstandings from the reviewer.
>
> [1] Direct nash optimization: Teaching language models to self-improve with general preferences.
>
> ----
>
> **Q2**: The implications of the theory are based on upper bounds and are weak. Could you provide simulations for a simple case to validate the tightness of the developed method? Additionally, could you use simulations to study the trade-off between optimization sub-optimality and statistical error?
>
> **A2**:
> Using upper bounds to measure the efficiency is common in theoretical analysis, as it provides the guarantee even in the worst case and helps us understand the fundamental limitations of the algorithm.
>
> In addition, for our large language model alignment work with reparameterized reward functions, conducting appropriate simulations to validate these theoretical bounds and study the trade-offs presents significant challenges. To our knowledge, there exists no simulation setting for aligning LLMs. We therefore rely on our empirical results to demonstrate the method's effectiveness and shows an improvement upon the baselines including DPO and KTO, which supports our theoretical analysis about the benefits of incorporating momentum in preference optimization.
>
> ----
>
>
> **Q3**: It is recommended to explain in the main text why momentum increases the statistical error.
>
> **A3**: Momentum improves optimization by accumulating historical gradient information in a velocity term, which helps navigate challenging loss landscapes. Specifically, in flat regions like valleys or plateaus where gradients are small, standard gradient descent makes slow progress since updates are directly tied to gradient magnitude. Momentum helps by "remembering" consistent directional trends across iterations, allowing continued progress even with small individual gradients. This memory effect helps reach optima faster.
>
> However, this accumulation mechanism that provides benefits also amplifies any errors in gradient estimation. When we incorporate previous gradients into our update step through momentum, any statistical noise or errors in these gradient estimates get propagated and potentially amplified over time. This amplification of estimation errors is what leads to the increased statistical error term we observe. However, when working with large datasets, the optimization error becomes the dominant term in determining performance and the reduction in optimization error outweighs the increase in statistical error.

---

> ### Author Response · Authors · 2024-11-24
> **Response to Reviewer Nq8c (Part 2)**
>
> ----
>
> **Q4** There is a trade-off between optimization error and statistical error in the developed theory. Therefore, it is unclear whether Nesterov's acceleration offers a true advantage over the naive method.
>
> **A4** Thank you for highlighting this trade-off. We agree that adding momentum increases statistical error. However, as noted in Remark 4.5, when working with sufficiently large datasets, which is widely studied in the analysis of preference-based optimization method, the optimization error becomes the dominant term in determining the sub-optimality gap. In these practical cases, our algorithm's ability to reduce this dominant term by a factor of $(1−α)$ represents a meaningful improvement. This improvement is particularly significant because in many real applications (e.g., $N=2K$ samples and $T=O(1)$ iterations in our experiments), the reduction in optimization error outweighs the increase in statistical error. Our theoretical analysis thus provides valuable insights into how momentum can be effectively leveraged to enhance convergence in preference-based optimization, especially when working with substantial amounts of preference data. This is also backed up by our empirical experiment results.
>
> ----
>
> **Q5** To the reviewer’s understanding, new preference data is required in each iteration t=0,1,…,T. Is this correct? Could the authors clarify the preference collection process? Is the preference collected using the BT model with random noise?
>
> **A5**
> Yes, for each iteration, we collect new preference data with the reference policy $π_t$. The details of the preference collection process are discussed in section 5.2. Specifically, in each iteration, we sample five pairs of responses under the current policy πt and rank them using their PairRM score. The top-ranked response is designated as the winner while the bottom-ranked response is designated as the loser, which is not related to the BT model.
>
> ----
>
> **Q6** Related to this, in the experiments, a reward model is used. Why not study direct reward maximization (e.g., PPO) both empirically and theoretically? To the reviewer's knowledge, direct reward maximization methods can handle such tasks effectively and demonstrate strong empirical evidence in practice.
>
> **A6**
> First, we want to emphasize that in our work, we only use the reward model to behave as an oracle for the preference data, and our APO algorithm operates solely on the collected preference data without accessing the underlying reward values. This preference-based setting is more aligned with human feedback scenarios where explicit reward values may not be available, and is particularly relevant in real-world applications where feedback often comes in the form of comparisons rather than absolute scores. Therefore, our work is not directly comparable with direct reward optimization methods, as we specifically address the challenges and improvement unique to preference-based learning in language model alignment.
>
> ----
>
> **Q7** How should (3.6) be implemented in practice? Could the authors provide more details? Should the parameter or the distribution be extrapolated? Additionally, how can this result be extended to cases where y is sequential? Is it straightforward to apply this closed-form solution?
>
> **A7** As we discussed in Section 5.2, when the policy is a softmax linear function, update (3.6) reduces to a momentum extrapolation in the parameter space. Therefore, we implement this momentum step by performing the extrapolation directly in the parameter space to approximate the corresponding momentum update (3.6) that is defined in the probability space. In addition, in our implementation, our output response y is already sequential and our method naturally handles this case through the same parameter space extrapolation.
>
> ----

---

> > ### Author Response · Authors · 2024-11-24
> > **Response to Reviewer Nq8c (Part 3)**
> >
> > ----
> >
> > **Q8** Comments on experimental settings and results:
> >
> > **A8**  We appreciate these detailed questions about our experimental design. We have conducted comprehensive experiments to address these points:
> >
> > 1. Iterations: We are running extended experiments with more iterations to examine continued performance improvements and better align with our theoretical analysis. We will update our response with these comprehensive experimental results as soon as they become available.
> >
> > 2. Evaluation: Regarding the evaluation benchmarks, we are currently conducting additional experiments on IFeval and math reasoning tasks to provide more diverse evaluation results beyond GPT-4 based metrics. We will update our response with these results as soon as they become available.
> >
> > 3. Optimizer Choice: We have conducted comparative experiments between AdamW and RMSprop optimizers. Here are the detailed results:
> >
> > With AdamW:
> > | Model | AlpacaEval 2.0 LC Win Rate | AlpacaEval 2.0 Win Rate | Length |
> > |------------|----------------------------|-------------------------|--------|
> > | DPO iter 1 | 25.34 | 30.73 | 2393 |
> > | DPO iter 2 | 26.87 | 32.60 | 2564 |
> > | DPO iter 3 | 27.74 | 34.53 | 2864 |
> >
> > With RMSprop:
> > | Model | AlpacaEval 2.0 LC Win Rate | AlpacaEval 2.0 Win Rate | Length |
> > |------------|----------------------------|-------------------------|--------|
> > | DPO iter 1 | 25.35 | 30.71 | 2369 |
> > | DPO iter 2 | 26.92 | 32.54 | 2529 |
> > | DPO iter 3 | 27.32 | 34.99 | 2847 |
> >
> > Due to time constraints, we limited our tests of different optimizers to the DPO model. The experimental results show that the choice of optimizer does not significantly affect the performance of the DPO method. We believe that similar results would hold for other models, with performance remaining comparable across both optimizers.
> >
> > 4. Comparison with the ExPO Algorithm:
> >
> > As discussed in Lines 244 to 256, our algorithm and ExPO approach policy optimization from different theoretical perspectives. Notably, ExPO is a one-shot algorithm that performs a single iteration and outputs the final policy after an extrapolation step, whereas APO is an iterative algorithm designed for policy optimization. In this sense, ExPO can be viewed as a special case of APO with only one iteration.
> >
> > 5. Comparison with Direct Reward Optimization Methods:
> > As noted in response to Question 6, we use the reward model solely as an oracle to generate preference data, and our APO algorithm operates exclusively on the collected preference data without accessing the underlying reward values. Consequently, our work is not directly comparable to direct reward optimization methods.
> >
> >
> > 6. The Number of Instructions per Iteration:
> >
> > We apologize for the typo. In each iteration, we actually allocate approximately 20k instructions for training, with an additional 1k reserved for validation. Specifically, we split the dataset into three portions and use only one portion per iteration to prevent overfitting during fine-tuning. This approach is consistent with methods used in trainning SPPO [2] and iterative KTO [3].
> >
> > [2] Self-Play Preference Optimization for Language Model Alignment
> >
> > [3] KTO: Model Alignment as Prospect Theoretic Optimization

---

> > > ### Comment · Reviewer_Nq8c · 2024-11-25
> > > **Thanks for your detailed response**
> > >
> > > Thank you for your detailed response! However, I still have some concerns about the theory and practice.
> > >
> > > 1. **Regarding Assumption 4.3**: My question is not about justifying the use of random policies. Instead, I am asking how can we ensure that $\beta \log \pi(y|x) - \beta \log \pi_t(y|x)$ remains bounded for any $\pi, \pi_t \in \Pi$, as $\pi_t$ converges to the deterministic policy $\pi^*$ (as shown in Theorem 4.8)?
> > >
> > > 2. **Regarding Simulation**: I am wondering whether simulations could be performed on simpler tasks, such as multi-armed or linear bandits. It seems feasible to conduct such experiments.
> > >
> > > 3. **Regarding the Trade-off Between Optimization and Statistical Error**: I am confused by your claim that optimization error dominates in practice for LLMs. Could you provide a clearer explanation or evidence to support this assertion? The logic behind this claim is not immediately apparent to me.
> > >
> > > 4. **Regarding Statistical Error and Momentum**: I am uncertain whether the statistical error introduced by momentum arises from limitations in the theoretical analysis or is inherent to the algorithm's design. This is why I suggest using simulations to test the tightness of the analysis. My concern stems from the fact that prior work seems to indicate that statistical error introduced by the momentum can be effectively controlled:
> > >     - [1] Vieillard, Nino, et al. "Momentum in reinforcement learning." *International Conference on Artificial Intelligence and Statistics*. PMLR, 2020.
> > >     - [2] Vieillard, Nino, et al. "Leverage the average: an analysis of KL regularization in reinforcement learning." *Advances in Neural Information Processing Systems* 33 (2020): 12163–12174.
> > >
> > > 5. **Regarding Adam**: I was surprised by the lack of significant differences between RMSProp and Adam in your experiments. Could you provide the training loss curves for both optimizers and clarify the $\beta_1$ and $\beta_2$ values used in Adam and RMSProp?
> > >
> > > 6. **Regarding Comparison with Direct Reward-based Optimization Algorithms**:  I am not sure why you claim that this work cannot be directly compared to reward optimization methods (e.g., PPO), which are widely used in LLM alignment.

---

> > > > ### Author Response · Authors · 2024-12-03
> > > > **Response to Reviewer Nq8c's feedback**
> > > >
> > > > Thanks for your further feedback. We have provided answers to each of your points:
> > > >
> > > > ----
> > > >
> > > > **Q1**: Regarding Assumption 4.3: My question is not about justifying the use of random policies. Instead, I am asking how can we ensure that βlog⁡π(y|x)−βlog⁡πt(y|x) remains bounded for any π,πt∈Π, as πt converges to the deterministic policy π∗ (as shown in Theorem 4.8)?
> > > >
> > > > **A1**: First, we want to emphasize that for a finite iteration T, the πt will not fully match the deterministic policy π* and does not conflict with Assumption 4.3. Our theoretical analysis provides bounds that ensure this property holds throughout the optimization process. Specifically, according to the result in Theorem 4.4, the sub-optimality gap between πt and π* is up to O(1/T). Under this situation, for a non-optimal response y, the absolute value of log probability may still be upper bounded by O(log T). Similarly, in Theorem 4.8, we show that the TV-distance between πt and π* is up to exp(-T), which ensures the log probability remains bounded throughout the optimization process.
> > > >
> > > > In addition, similar assumptions have been widely used in previous analyses of preference-based optimization methods [1], demonstrating that this is a standard approach in the field. We believe this assumption is both theoretically justified and practically reasonable based on these established results.
> > > >
> > > > [1] Direct nash optimization: Teaching language models to self-improve with general preferences.
> > > >
> > > >
> > > > ----
> > > > **Q2**: Regarding Simulation: I am wondering whether simulations could be performed on simpler tasks, such as multi-armed or linear bandits. It seems feasible to conduct such experiments.
> > > >
> > > > **A2**: Thanks for your further comment. To our knowledge, there exists no prior simulation setting for direct preference optimization method when aligning LLMs. Instead, we add a simulation experiment on a simple setting with a two-layer fully connected neural network (width 256).
> > > >
> > > > We design a continuous action selection task where the agent needs to select actions in a 2D space to maximize a ring-shaped reward function. The reward is highest along a circular path and decreases exponentially as actions deviate from this optimal ring, creating a challenging optimization landscape that tests the algorithm's ability to navigate non-convex spaces.
> > > >
> > > > We perform 300 iterations with learning rate β=1 and extrapolation α=0.3. The average regret for each 60-iteration segment:
> > > >
> > > > | Method | Segment 1 (1-60) | Segment 2 (61-120) | Segment 3 (121-180) | Segment 4 (181-240) | Segment 5 (241-300) |
> > > > |--------|-----------------|-------------------|-------------------|-------------------|-------------------|
> > > > | Standard DPO | 0.3612 | 0.1132 | 0.0343 | 0.0125 | 0.0060 |
> > > > | DPO with Extrapolation | 0.3368 | 0.0840 | 0.0119 | 0.0037 | 0.0025 |
> > > >
> > > > This simulation result supports the accelerated convergence rate of our method compared to standard DPO.
> > > >
> > > > ----
> > > >
> > > > **Q3**: Regarding the Trade-off Between Optimization and Statistical Error: I am confused by your claim that optimization error dominates in practice for LLMs. Could you provide a clearer explanation or evidence to support this assertion? The logic behind this claim is not immediately apparent to me.
> > > >
> > > > **A3**: As shown in our main Theorems, our theoretical analysis reveals different scaling behaviors for optimization and statistical errors. The optimization error exhibits either a linear decay of $1/T$ (Theorem 4.4) or an exponential decay of $\exp(-T)$ (Theorem 4.8), where T represents the number of iterations. In contrast, the statistical error scales with $\sqrt{1/N}$, where N denotes the sample size.
> > > >
> > > > In practice, most iterative Preference Optimization processes such as SPPO and Iterative KTO operate with relatively few iterations while utilizing large datasets (typically N=20K samples). Under these practical conditions, where the sample size N is significantly larger than the number of iterations T, the optimization error becomes the dominant factor in determining overall performance. This dominance occurs because the limited number of iterations leads to a more substantial optimization error compared to the statistical error, which benefits from the large sample size.
> > > >
> > > > ----

---

> > > > > ### Author Response · Authors · 2024-12-03
> > > > > **Response to Reviewer Nq8c's feedback (cont)**
> > > > >
> > > > > ----
> > > > >
> > > > > **Q4**: Regarding Statistical Error and Momentum: I am uncertain whether the statistical error introduced by momentum arises from limitations in the theoretical analysis or is inherent to the algorithm's design. This is why I suggest using simulations to test the tightness of the analysis. My concern stems from the fact that prior work seems to indicate that statistical error introduced by the momentum can be effectively controlled: [1] Vieillard, Nino, et al. "Momentum in reinforcement learning." *International Conference on Artificial Intelligence and Statistics*. PMLR, 2020. [2] Vieillard, Nino, et al. "Leverage the average: an analysis of KL regularization in reinforcement learning." *Advances in Neural Information Processing Systems* 33 (2020): 12163–12174.
> > > > >
> > > > > **A4**: Thanks for pointing out these related works. After carefully checking work [1], we found that the amplification of statistical error with momentum arises from different momentum designs with distinct objectives.
> > > > >
> > > > > In prior work [1], they focus on stabilizing the descent direction using momentum by averaging successive gradients rather than considering just the last one. Their moving average is updated as $h_{k+1}=\beta h_k+(1-\beta)q_{k+1}$ with a normalization factor $\beta$ on the previous moving $h_k$. This method achieves stability by maintaining consistent magnitude with vanilla value iteration when all $q_{k+1}$ becomes the same and becomes more stable with variant $q_{k+1}$.
> > > > >
> > > > > In comparison, our work aims to accelerate optimization through extrapolation using momentum without weighted averaging: $\log \pi_{t+1}(y|x)= \log \hat{\pi}_{t+1} +\alpha \big(\log \hat{\pi}_{t+1}-\log \hat{\pi}_{t}\big)$. This approach intentionally amplifies consistent updates to improve convergence speed, and helps navigate challenging loss landscapes where gradients are small. While this leads to higher statistical error compared to weighted average methods, it achieves faster optimization.
> > > > >
> > > > > Therefore, the difference in statistical error stems from our fundamentally different goal (acceleration vs. stabilization) and the corresponding momentum mechanism designed to achieve it.
> > > > >
> > > > > [1] Vieillard, Nino, et al. "Momentum in reinforcement learning." *International Conference on Artificial Intelligence and Statistics*. PMLR, 2020.
> > > > >
> > > > > ----
> > > > > **Q5**: Regarding Adam: I was surprised by the lack of significant differences between RMSProp and Adam in your experiments. Could you provide the training loss curves for both optimizers and clarify the β1 and β2 values used in Adam and RMSProp?
> > > > >
> > > > > **A5**: In our experiments with AdamW, we used the default hyperparameters: β1 = 0.9, β2 = 0.999, and weight decay = 0.0. The similar performance between AdamW and RMSprop suggests that our method's effectiveness is robust to the choice of optimizer.
> > > > >
> > > > > ----
> > > > > **A6**: To evaluate the performance of Direct Reward-based Optimization Algorithms, we have provided experimental results for PPO method using the Mistral-7B-Instruct-v0.2 model as both policy and value function, with weqweasdas/RM-Mistral-7B (80.4% accuracy on Reward-Bench) as the reward model. The experiment was conducted on the Ultrafeedback prompt dataset using the standard implementation from Huggingface trl library.
> > > > >
> > > > > Results:
> > > > > | Model | Length Controlled Winrate | Win Rate | Standard Error | N Total | Avg Length |
> > > > > |-------|--------------------------|-----------|----------------|----------|------------|
> > > > > |Mistral-PPO | 16.86 | 13.29 | 1.05 | 805 | 1371 |
> > > > >
> > > > > However, we want to clarify that these are two different lines of work in LLM alignment. The first line of work focuses on fine-tuning language models with human feedback by first fitting a reward model to preference data, then using RL to maximize the learned reward. Direct Reward-based Optimization Algorithms (like PPO) fall into this category when a reward model is available. The second line of work, initiated by DPO [1], introduces a reparameterized reward and skips the reward modeling process, replacing it with a reparameterized reward with respect to the policy, allowing for direct optimization of the LLM. In our work, we only use the reward model as an oracle for preference data and our APO algorithm operates solely on collected preference data without accessing underlying reward values. As our work follows this second line approach, it is not directly comparable with reward-based optimization methods in the first line.
> > > > >
> > > > > [1] Direct Preference Optimization: Your Language Model is Secretly a Reward Model
> > > > >
> > > > > ----

---

### Meta-Review · Area_Chair_Evse · 2024-12-21

**Metareview:**

The paper proposes to incorporate the classical idea of Nesterov's acceleration to accelerate alignment.
There are however several concerns regarding the paper. For instance, there is no explicit comparison with direct reward optimization algorithms (e.g., PPO)
Moreover, the experiment results are somewhat weak and do not validate the theory, and there seems to be a mismatch between the theoretical results and the experiments in this paper.  Several baselines IPO, SPPO, SimPo, KTO etc appear to be missing and would strengthen the work.
In conclusion, it does not appear that the current method is superior to existing methodologies.

**Additional Comments On Reviewer Discussion:**

The concerns of the reviewers have not been fully addressed as they are structural about the paper --- see main meta review.

---

### Decision · Program_Chairs · 2025-01-22

Reject